# An essential dual-function complex mediates erythrocyte invasion and channel-mediated nutrient uptake in malaria parasites

Daisuke Ito, Marc A Schureck, Sanjay A Desai*

Laboratory of Malaria and Vector Research, NIAID, National Institutes of Health, Rockville, United States

**Abstract** Malaria parasites evade immune detection by growth and replication within erythrocytes. After erythrocyte invasion, the intracellular pathogen must increase host cell uptake of nutrients from plasma. Here, we report that the parasite-encoded RhopH complex contributes to both invasion and channel-mediated nutrient uptake. As *rhoph2* and *rhoph3* gene knockouts were not viable in the human *P. falciparum* pathogen, we used conditional knockdowns to determine that the encoded proteins are essential and to identify their stage-specific functions. We exclude presumed roles for RhopH2 and CLAG3 in erythrocyte invasion but implicate a RhopH3 contribution either through ligand-receptor interactions or subsequent parasite internalization. These proteins then traffic via an export translocon to the host membrane, where they form a nutrient channel. Knockdown of either RhopH2 or RhopH3 disrupts the entire complex, interfering with organellar targeting and subsequent trafficking. Therapies targeting this complex should attack the pathogen at two critical points in its cycle.

*For correspondence: sdesai@niaid.nih.gov

## Introduction

Malaria parasites are sophisticated single-celled organisms that infect erythrocytes of many vertebrate animals; in humans, the virulent *Plasmodium falciparum* pathogen remains a public health priority because of high mortality rates and global economic costs. These parasites have evolved novel proteins, some of which form multi-subunit complexes, to mediate host cell invasion and to enable intracellular survival.

One multi-protein complex, the RhopH complex, was identified in early proteomic studies of invasive merozoites (*Campbell et al., 1984*; *Holder et al., 1985*), but remains poorly characterized. Its name recognizes that it is a high-molecular weight complex localizing to rhoptries, which are specialized organelles at the apical end of the merozoite (*Holder et al., 1985*; *Cooper et al., 1988*; *Kaneko et al., 2001*; *Ling et al., 2003*). Each of the three member proteins, termed CLAG (or RhopH1), RhopH2, and RhopH3, are synthesized in mature intracellular parasites known as schizonts (*Kaneko et al., 2001*, *2005*). Upon egress, these proteins are secreted, along with other rhoptry contents, onto the erythrocyte targeted for invasion. It has therefore been assumed that the RhopH complex serves a role in invasion; because a fraction of this complex is transferred to the new host cell, another proposal has been that these proteins contribute to formation of the parasitophorous vacuole that surrounds the internalized parasite (*Kaneko, 2007*; *Ocampo et al., 2005*; *Ranjan et al., 2011*; *Cortés et al., 2007*).

Surprisingly, genetic mapping in *P. falciparum* subsequently implicated the CLAG paralogs on chromosome 3, CLAG3.1 and CLAG3.2, in the formation of the plasmodial surface anion channel

**eLife digest** The parasites that cause malaria in humans and other animals infect and live inside red blood cells to escape attack by their hosts' immune systems. Malaria parasites grow and multiply in red blood cells before bursting out and invading new red blood cells. To fuel this growth, the parasite needs access to sugars and other nutrients that are found outside in the bloodstream. Malaria parasites achieve this by inserting some of their own proteins into the membrane of the red blood cell to form an unusual channel that allows the nutrients to enter the cell.

A parasite protein called CLAG3 (also known as RhopH1) is involved in formation of the unusual nutrient channel. Unlike most other proteins, malaria parasites make the CLAG3 protein while they are inside one cell and release it later when they invade a new red blood cell. The CLAG3 protein also binds to two other parasite proteins, called RhopH2 and RhopH3, to form a larger protein complex. However, it was not known what roles these other proteins played, or why the complex was made in the preceding red blood cell.

Ito et al. have now addressed these unknowns by editing the genes of the parasite that causes the most dangerous form of malaria in people, a parasite called *Plasmodium falciparum*. These experiments revealed that the parasites could still invade host cells as normal if they lost CLAG3 and RhopH2. This suggests, that contrary to what was expected, CLAG3 and RhopH2 are not needed for the invasion process. Instead, the experiments revealed that RhopH3 serves a major role in invasion, either by helping the parasite to interact with or enter the new red blood cell. After the parasite has invaded the cell, this complex of three proteins is shuttled to the red blood cell's membrane, where it inserts to help form the nutrient channel.

The findings of Ito et al. reveal that one protein complex serves two unrelated but essential roles at different locations and time points in the life cycle of a malaria parasite. Since a parasite will not survive if it cannot enter a host cell and obtain nutrients, interfering with these processes by targeting this protein complex could lead to new therapies against malaria in the future.

(PSAC; *Nguitragool et al., 2011*), a nutrient and ion channel at the host cell membrane (*Desai et al., 2000*). Subsequent studies with transport mutants and specific inhibitors have strengthened the evidence for CLAG3 and possibly CLAG2 in channel activity (*Pillai et al., 2012*; *Sharma et al., 2013*; *Mira-Martinez et al., 2013*; *Sharma et al., 2015*). Although conservation of both the RhopH complex and PSAC activity in all examined *Plasmodia* spp. suggests a causal link (*Lisk and Desai, 2005*), the precise contribution of these proteins to either the activation of channels or the formation of the nutrient pore is unclear (*Desai, 2012*). Growth defects in a reported CLAG3 knockout are consistent with a role in nutrient uptake, but could also result from the loss of other activities (*Comeaux et al., 2011*). The multiple and seemingly conflicting proposals for the biological roles of member proteins in this complex have yet to be parsed out (*Gupta et al., 2015*).

In contrast to the *clag* gene family, *rhoph2* and *rhoph3* are present as single copy genes in each *Plasmodium* species. The encoded RhopH2 and RhopH3 proteins remain poorly characterized. Here, we used parasite DNA transfection to edit these genes and to examine their roles throughout the parasite life cycle. Our studies show that both proteins are essential for parasite survival. RhopH3, but not RhopH2 or CLAG3, contributes to erythrocyte invasion. Because the formation and functional properties of the parasitophorous vacuole are preserved in our knockdowns, these three proteins do not appear to serve long-assumed roles in vacuole biogenesis. Instead, organic solute uptake and electrophysiological studies reveal that both RhopH2 and RhopH3 are required for successful PSAC formation; loss of channel activity leads to rapid parasite demise. These critical functions and the elaborate stage-specific trafficking we report makes the RhopH complex an attractive therapeutic target.

## Results

### RhopH members are membrane-associated proteins that are transferred to erythrocytes during invasion

Each member of the RhopH complex is membrane-associated but can be partially extracted by $Na_2CO_3$ at pH 11 (*Figure 1A*), suggesting distinct pools are peripheral and integral to membranes. While pronase E treatment of infected cells confirmed CLAG3 proteolysis at a variant surface-exposed loop (*Figure 1A*; *Nguitragool et al., 2014*), RhopH2 and RhopH3 were not cleaved, suggesting that they are not exposed at the host membrane. The resistance of RhopH2 and RhopH3 to pronase E, the relative absence of polymorphisms in the genes that encode these proteins in *P. falciparum* clones (*Iriko et al., 2008*), and CLAG3 proteolysis at a variant motif are all consistent with polymorphism at surface-exposed epitopes; such polymorphisms are presumably selected by host immune attack.

Immunofluorescence microscopy confirmed packaging into rhoptries within schizont-infected cells (*Figure 1B*). Imaging of maturing trophozoite-stage parasites also showed that these proteins traffic to the host membrane (*Figure 1C*), consistent with earlier localization studies (*Vincensini et al., 2008*). Because each gene is transcribed and translated only within schizonts, the protein found in these maturing trophozoites reflects transfer from the merozoite during invasion (*Lustigman et al., 1988*; *Ling et al., 2004*). To quantify this transfer, we performed stage-specific immunoblotting and found that about half of each protein was lost to the culture supernatant, where it associated with the ultracentrifugation pellet ('spent', *Figure 1D*). After invasion, each protein's abundance remained relatively constant throughout parasite maturation from ring to trophozoite stages (*Figure 1D*), consistent with biochemically stable proteins that do not undergo further synthesis until the schizont stage.

### CRISPR-Cas9 targeting yields knockdowns, but not knockouts

We next used transfection to explore essentiality and possible functions of RhopH2 and RhopH3 (*Figure 1E*). Neither gene could be disrupted ('KO', *Figure 1F*) by transfection with two separate single guide RNAs (sgRNAs) designed from each gene's 5′ end (*Supplementary file 1*). At the same time, these sgRNAs yielded robust integration of a codon-optimized gene fragment that preserves the encoded protein sequences ('con'), indicating effective targeting of these genomic sites and tolerance for synonymous DNA changes. These findings suggest that both RhopH2 and RhopH3 serve essential functions.

To identify possible roles, we used two conditional knockdown approaches and began with the *glmS* ribozyme (*Prommana et al., 2013*), a 160-bp RNA element that is cleaved upon interaction with glucosamine (GlcN). We introduced *glmS* into each gene's 3′ UTR using specific sgRNAs (*Figure 2A* and *Supplementary file 1*); limiting dilution clones, *R2glmS* and *R3glmS*, carried the riboswitch without residual wild-type sequence. Both parasites exhibited unchanged growth in the absence of GlcN (data not shown), consistent with a benign distal modification of each gene.

### Knockdowns co-deplete CLAG3 and reveal that only RhopH3 contributes to invasion

To evaluate riboswitch effectiveness, we added GlcN to parasite cultures throughout *rhoph* gene transcription (addition at trophozoite stage and continued through egress and reinvasion, *Figure 2B*). Immunoblotting revealed that this treatment reduced each targeted protein to essentially undetectable levels in schizont-infected cells (*Figure 2C*). This suppressor had no effect on RAP1, an unrelated membrane-associated rhoptry protein (*Baldi et al., 2000*), or on these proteins in untransfected parasites, indicating specific action at the introduced *glmS* riboswitch.

Remarkably, GlcN treatment of either *R2glmS* or *R3glmS* abolished not only the targeted gene product but also CLAG3, a third better-characterized member of the RhopH complex. Indirect immunofluorescence microscopy confirmed this surprising finding and provided additional insights (*Figure 2D*). While RhopH2 was still detected in the *R3glmS* knockdown, it did not fully colocalize with RAP1 but instead had a more diffuse distribution; this suggests that RhopH2 may be retained in the parasite endoplasmic reticulum if it fails to interact with RhopH3. By contrast, RhopH3 appeared to traffic normally in GlcN-treated *R2glmS* parasites, as indicated by its apical colocalization with

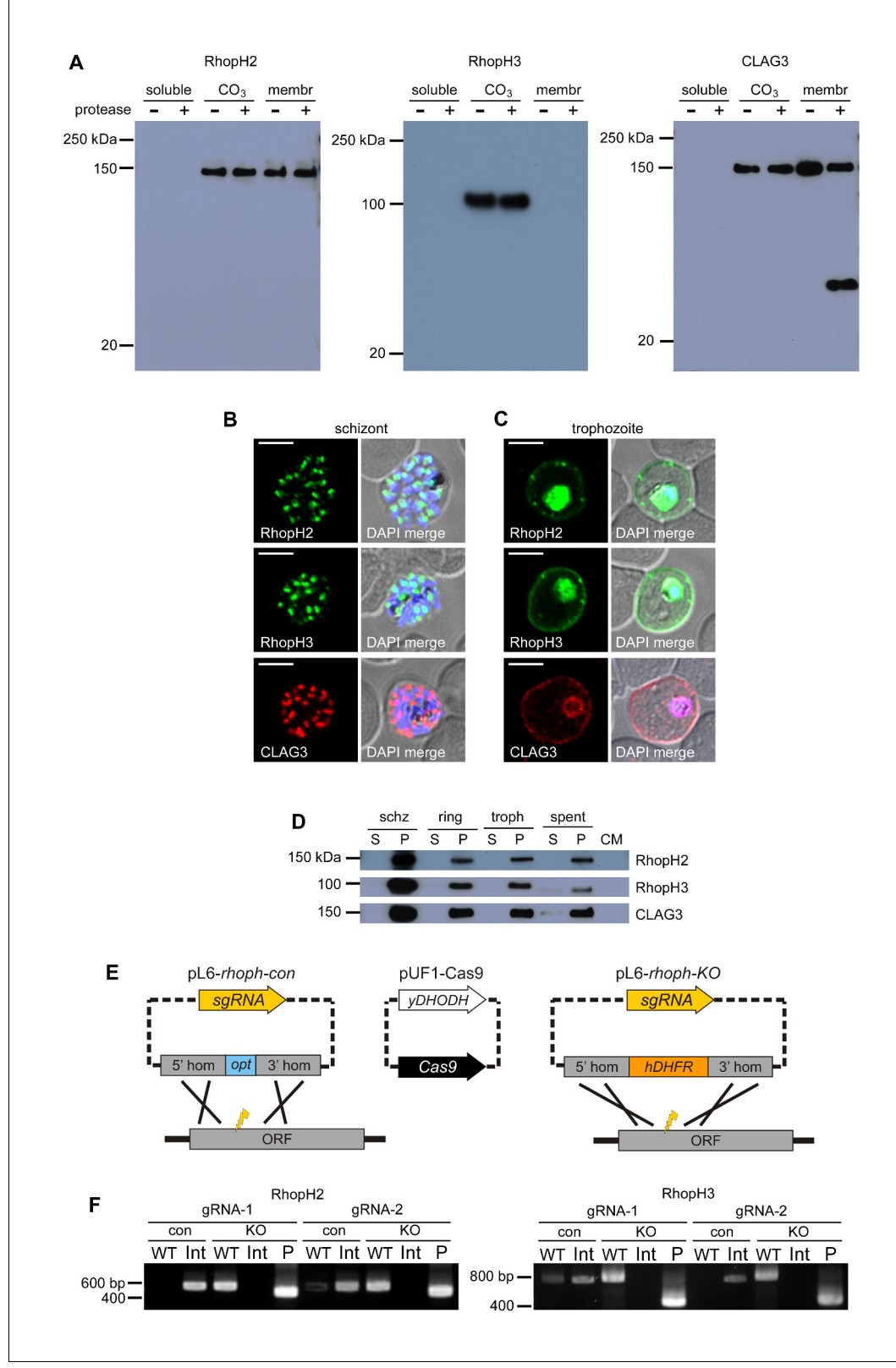

**Figure 1.** RhopH2 and RhopH3 are membrane-associated proteins whose genes cannot be disrupted. (**A**) Immunoblots showing that members of this complex are membrane-associated, but extractable by $Na_2CO_3$ ($CO_3$); membr, integral membrane pool resistant to $CO_3$. Only CLAG3 is cleaved by external pronase E, as indicated by an additional ~35 kDa band. (**B**) Immunofluorescent antibody (IFA) imaging of mature schizont-infected cells with

*Figure 1 continued on next page*

*Figure 1 continued*

indicated antibodies. Punctate labeling indicates that these proteins localize to rhoptry organelles in daughter merozoites. Scale bar, 5 μm. (**C**) IFA of trophozoite-infected cells, showing that each protein localizes to the host membrane and to the vacuole surrounding the intracellular parasite. Scale bar, 5 μm. (**D**) Matched-loading immunoblots from parasite stages shows production in schizonts (schz), incomplete delivery to invaded erythrocytes (ring- and trophozoite-infected cells), and release into culture supernatant (spent). CM, culture medium prior to cultivation is shown as a negative control. In each sample, proteins remain associated with membrane pellets (P) and are not soluble (S). (**E**) Schematic illustrates two plasmid transfection to produce knockouts (pL6-*rhoph-KO*), or transfection control that verifies sgRNA efficacy through repair with a codon-optimized version encoding an unmodified protein (pL6-*rhoph-con*). *Streptococcus pyogenes* Cas9 is expressed from pUF1-Cas9 (**Ghorbal et al., 2014**). (**F**) Ethidium-stained gel showing PCR-confirmed integration (Int) of positive control transfections (con) for each sgRNA on both genes, but retention of a wild-type site (WT) and pL6 episomes (P) in knockout transfections (KO).

RAP1 in merozoites. Thus, CLAG3, RhopH2 and RhopH3 assemble into a complex either during or immediately after translation. RhopH3 reaches rhoptries in the absence of RhopH2, but the other two components are either rendered unstable or mis-trafficked in each knockdown. The unchanged intensity and distribution of RAP1 in both knockdowns excludes essential roles of RhopH2 or RhopH3 in trafficking unassociated proteins or in rhoptry formation.

Previous studies have reported erythrocyte binding for each RhopH complex member and have proposed roles in host cell invasion (**Sam-Yellowe, 1992**; **Ocampo et al., 2005**; **Baldwin et al., 2014**). While RhopH3 antibodies also inhibit invasion, this protein's interactions with other RhopH members and possibly with other ligands prevent clear assignment of invasion roles (**Ranjan et al., 2011**). Interpretation is also complicated by antibody cross-reactivity, a well-established problem for parasite antigens (**Holmquist et al., 1988**; **Udomsangpetch et al., 1989**). We therefore examined the effect of knockdowns on this critical process. Dose-response studies revealed that GlcN had no measurable effect on egress and erythrocyte invasion by wild-type or *R2glmS* parasites (**Figure 2E**). Because the *R2glmS* knockdown co-depletes both RhopH2 and CLAG3 (**Figure 2C and D**), fully preserved invasion in this parasite definitively refutes proposals that either of these proteins contributes to egress or invasion.

By contrast, *R3glmS* exhibited a dose-dependent effect on invasion, with 4 mM GlcN decreasing the number of ring-infected cells by 52% ± 3.0% (p=0.002, n = 3). This invasion defect did not result from failed maturation or reduced merozoite number because RAP1 labeling indicated that these were not compromised (**Figure 2D**). Giemsa staining revealed extracellular merozoites (red arrows, **Figure 2F**), excluding effects on host cell rupture or merozoite egress. These findings restrict RhopH3's precise contribution to a role in either host cell interaction or subsequent parasite internalization. Notably, the observed CLAG3 co-depletion mandated examination of both knockdowns and the serendipitously intact RhopH3 trafficking in *R2glmS* to exclude contributions from RhopH2 and CLAG3, and to determine that RhopH3 is the sole member participating in invasion.

## Subsequent transfer to new host cells requires an intact complex

We continued cultivation of GlcN-induced knockdowns and observed that these parasites progressed normally to the next trophozoite stage (**Figure 3A**). Harvest and immunoblotting at this stage revealed complete loss, not only of the targeted proteins, but also of each other RhopH complex member (**Figure 3B**), a finding we confirmed with immunofluorescence assays (IFAs, **Figure 3C**). Interestingly, even RhopH3, which retained its rhoptry trafficking and invasion-associated functions in the RhopH2 knockdown (**Figure 2D and E**), failed to transfer to new erythrocytes in GlcN-treated *R2glmS*. As GlcN was without effect in untransfected parasites (**Figure 3—figure supplement 1A**), these findings indicate that these proteins require interactions with their cognate partners for successful transfer to the next erythrocyte at the time of invasion. They argue against transfer by simple entrapment of rhoptry proteins in the nascent parasitophorous vacuole during invasion (**Kats et al., 2008**).

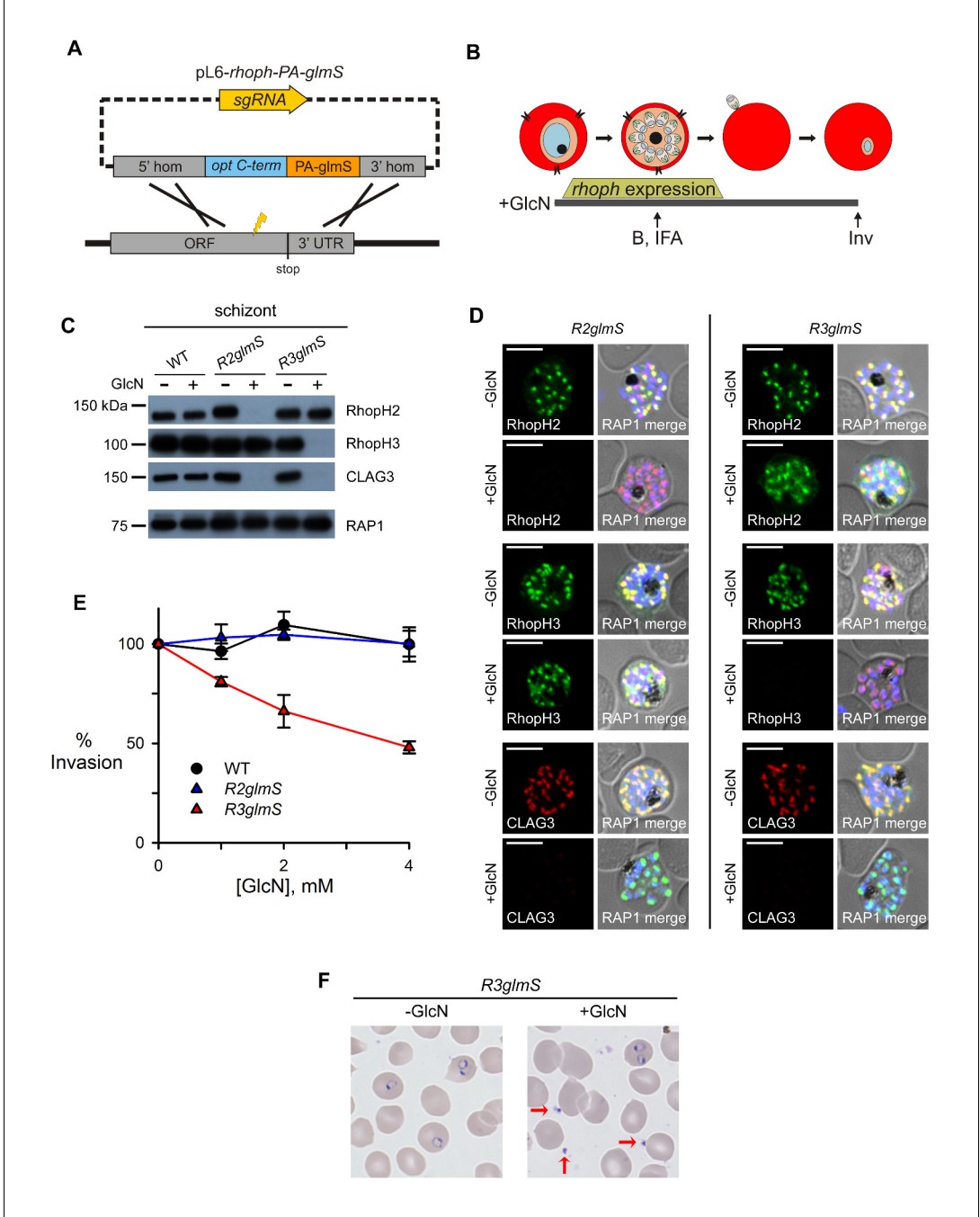

**Figure 2.** Assembly-dependent trafficking in merozoites and a role in host cell invasion for RhopH3 alone. (**A**) Strategy for the introduction of the PA epitope tag followed by a stop codon and a *glmS* riboswitch at the 3' end of *rhoph* genes. (**B**) Schematic showing GlcN exposure timed to cover the expression of *rhoph* genes and the subsequent harvest of cells for immunoblotting (B), indirect immunofluorescence assays (IFA), and invasion studies (Inv). (**C**) Immunoblots of matched lysates from schizont-stage parasites with or without exposure to 4 mM GlcN. Knockdown eliminates targeted gene product and also CLAG3. RAP1, an unassociated rhoptry protein, is preserved. (**D**) IFA of indicated proteins in *R2glmS* and *R3glmS* at schizont stage; the colocalization of a protein with RAP1 and apical puncta in daughter merozoites indicates rhoptry trafficking. Scale bars, 5 μm. (**E**) GlcN dose responses (mean ± S.E.M.) for merozoite invasion by each parasite. (**F**) Giemsa-stained micrographs of *R3glmS* showing extracellular merozoites (red arrows) after GlcN treatment, but not in the untreated control. Immature ring-infected erythrocytes are also apparent.

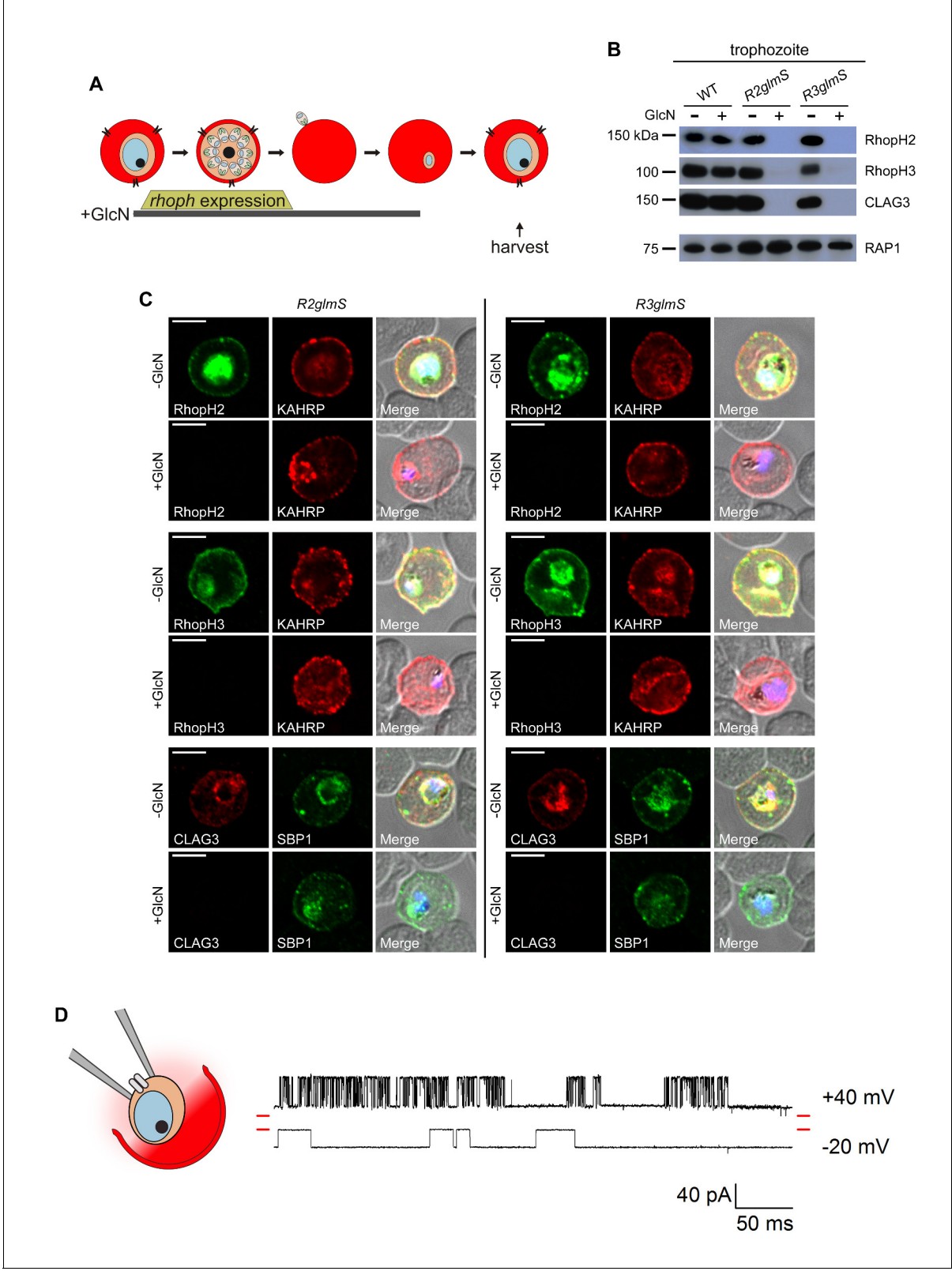

**Figure 3.** Transfer of intact complexes to new host cells and no role for RhopH proteins in vacuolar function. (**A**) Schematic showing timeline for GlcN exposure, *rhoph* gene expression, and subsequent harvest of trophozoite-infected cells. (**B**) Immunoblots of matched lysates from trophozoite-stage parasites with or without exposure to 4 mM GlcN. Both knockdown parasites lack all three associated proteins, indicating that only intact complexes are transferred during host cell invasion. RAP1 does not depend on RhopH proteins for its transfer. (**C**) IFA of indicated proteins in *R2glmS* and *R3glmS* at

*Figure 3 continued*

the trophozoite stage. Each member of the complex is undetectable in GlcN-treated cells, but KAHRP and SBP1 are exported to the host compartment normally. Scale bars, 5 μm. (D) Single-channel patch-clamp recordings on the PVM of *R3glmS* after cultivation with 4 mM GlcN, showing stochastic transitions of the large conductance channel at this membrane. The RhopH complex is not required for this channel activity. The membrane potential at the patch (negative of the pipette potential) is indicated to the right of each trace; red dashes at both ends of traces correspond to the closed channel levels. 460 ms of recording is shown at each membrane potential. The schematic on the left shows a patch-clamp pipette sealed on the PVM after parasite extraction from a host erythrocyte.

The following figure supplement is available for figure 3:

**Figure supplement 1.** GlcN affects neither RhopH complex proteins in untransfected parasites nor PVM channel biophysical properties.

Whether RhopH2 and/or RhopH3 can be transferred to the next erythrocyte in the absence of CLAG proteins remains unclear because it is currently not possible to produce a parasite lacking all five *clag* paralogs in the *P. falciparum* genome.

## Activities at the parasitophorous vacuole are unaffected

The unaffected morphology of trophozoite-stage parasites, despite undetectable RhopH protein levels (*Figure 3B and C*), suggests normal formation of the parasitophorous vacuole (PV) and its surrounding membrane (PVM). Export of two parasite proteins, KAHRP and SBP1, into the host cytosol was also unchanged (*Figure 3C*), suggesting that neither RhopH member contributes to the protein export translocon at this membrane (*de Koning-Ward et al., 2009*; *Beck et al., 2014*).

Cell-attached patch-clamp of the PVM in the *R3glmS* knockdown revealed that loss of the RhopH complex does not adversely affect a large conductance channel on this membrane (*Desai et al., 1993*; *Desai and Rosenberg, 1997*). PVM patch-clamp depends on extraction of individual parasites from their host cell while preserving vacuole integrity (*Figure 3D* schematic); previous studies using this method have revealed a single broad selectivity ion channel, as evidenced by stochastic opening and closing events at imposed membrane potentials. The *R3glmS* knockdown exhibited single-channel conductance, sub-conductance states, and channel voltage dependence matching those reported previously (*Figure 3D* and *Figure 3—figure supplement 1B–D*). As knockdown abolishes PSAC activity at the host erythrocyte membrane (described below), these observations implicate a PVM channel molecular composition distinct from that of PSAC. On the basis of the preserved structure and function at this membrane, our findings exclude previously proposed roles for RhopH members in PV biogenesis and suggest, instead, that these proteins only transit through this compartment.

## The RhopH complex is required for nutrient channels on the host membrane

CLAG3 determines PSAC activity at the host membrane by unknown mechanisms. We therefore examined infected erythrocyte permeability and began with kinetic measurements of osmotic lysis in sorbitol, a sugar alcohol with primary uptake via PSAC (*Pillai et al., 2010*). This kinetic assay provides quantitative estimates of organic solute permeability and is specific for PSAC, as confirmed by inhibitor studies that match single channel recordings (*Pillai et al., 2010*).

Using GlcN-treated parasites cultivated to the trophozoite stage, sorbitol uptake was drastically reduced in both *R2glmS* and *R3glmS* (p<0.001), but unaffected in the wild-type control (*Figure 4A–C*). Interestingly, dose-response studies found $EC_{50}$ values of 0.26 ± 0.02 and 0.40 ± 0.02 mM for GlcN, respectively (*Figure 4D*); these values are lower than those observed with GFP and DHFR reporter assays (*Prommana et al., 2013*). This greater than expected efficacy, as well as the large proportion of cells that are fully refractory to osmotic lysis (*Figure 4E*), may reflect either the cumulative effect of losing all three RhopH complex members or a higher-order stoichiometry for these proteins in PSAC formation. There was also a uniform reduction in the permeabilities of amino acid, sugar alcohol, and organic cation fluxes upon knockdown (*Figure 4—figure supplement 1A*). These findings suggest that both knockdown parasites have a reduced number of channels with unaltered solute selectivities; this contrasts with previously characterized PSAC mutants, which exhibit unequal

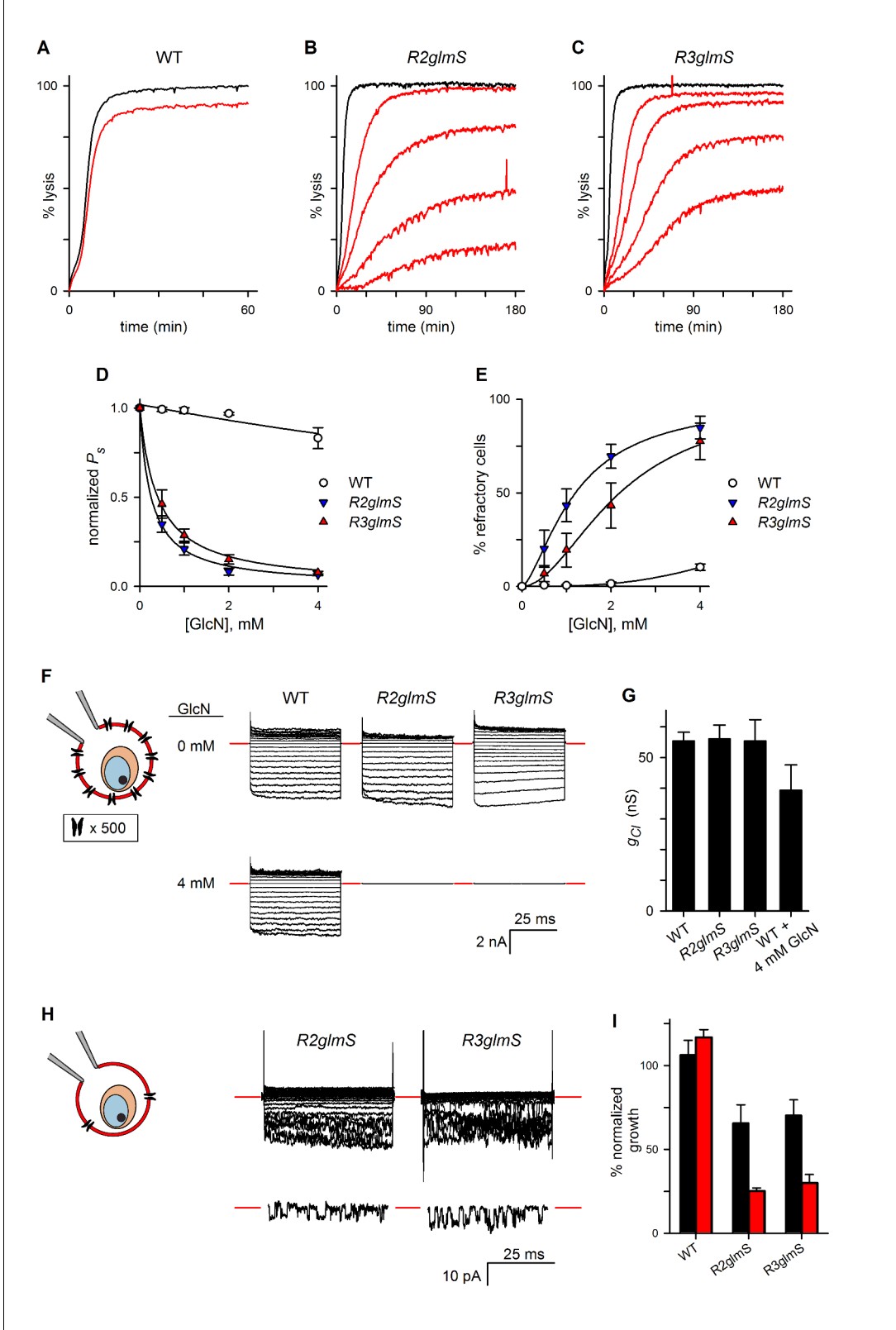

**Figure 4.** RhopH complex abundance correlates with PSAC activity. (**A**) Osmotic lysis kinetics resulting from sorbitol uptake by wild-type parasites with or without exposure to 4 mM GlcN (red and black traces, respectively). (**B–C**) Lysis kinetics for transfectant parasites without (black) or with 0.5, 1, 2, or 4 mM GlcN (top to bottom red traces, respectively). Note the extended time scales for osmotic lysis. (**D**) Normalized sorbitol permeabilities (mean ± S.E. M.) for each parasite at indicated GlcN concentrations. Solid lines represent best fits to $y = a/(1 + (x/K_{0.5}))$. (**E**) Percentage of cells refractory to osmotic

*Figure 4 continued on next page*

*Figure 4 continued*

lysis (mean ± S.E.M.) at each [GlcN]. Solid lines represent best fits to a two-parameter logistic curve. (**F**) Whole-cell currents from individual trophozoite-infected erythrocytes, presented as the ensemble responses to membrane voltages ($V_m$) between −100 and +100 mV in 10 mV increments. 4 mM GlcN specifically abolishes currents on knockdown parasites. Red dashes indicate zero current levels. The schematic on the left shows the whole-cell patch-clamp configuration; the recording pipette measures currents resulting from ion flow through channels on the cell surface; the measured currents correspond to ~4000 channels on infected cells not subjected to GlcN knockdown. (**G**) Chord conductances ($g_{Cl}$) (mean ± S.E.M.) for each parasite without GlcN, calculated from whole-cell currents between $V_m$ of 0 and −100 mV. Statistics were calculated from up to 15 cells each. The value for wild-type cells after 4 mM GlcN treatment is also shown. (**H**) Whole-cell ensemble currents for transfectants cultivated with 4 mM GlcN, taken from panel (**F**) but presented at an increased gain. Single traces beneath each ensemble show stochastic transitions that are due to openings of individual channels detected in the whole-cell configuration; $V_m$, −90 and −70 mV for traces shown at bottom in *R2glmS* and *R3glmS*, respectively. These transitions and the reduced ensemble amplitudes indicate a marked reduction in PSAC copy number. Horizontal and vertical scale bars represent 25 ms and 10 pA, respectively, for all traces. The schematic on the left shows the whole-cell patch-clamp configuration and a reduced number of channels. (**I**) Parasite growth (mean ± S.E.M.) over 72 hr in standard medium or PGIM (black and red bars, respectively) after exposure to 1 mM GlcN. For each parasite, growth was normalized to 100% for matched controls using the same parasite and culture medium without GlcN treatment.

The following figure supplement is available for figure 4:

**Figure supplement 1.** Organic solute permeabilities and cell-attached patch-clamp of wild-type and *glmS* knockdown parasites.

reductions in solute permeabilities as a consequence of altered single-channel properties (*Hill et al., 2007*; *Lisk et al., 2008*).

In the light of these findings, we performed whole-cell patch-clamp to quantify total ion flux across the host membrane of individual infected erythrocytes (*Desai, 2012*). Without GlcN, wild-type and transfectant parasites yielded indistinguishable Cl⁻conductances with characteristic inward-rectifying behavior (apparent as larger downward deflections in an ensemble depiction of currents, *Figure 4F*). 4 mM GlcN produced a statistically insignificant reduction in Cl⁻ conductance for wild-type cells ($g_{Cl}$, *Figure 4G*, n = 4–15 cells each). By contrast, the same treatment abolished measurable PSAC-mediated currents on nearly all cells infected with *R2glmS* or *R3glmS* (n = 8–16 cells each), consistent with a large fraction of cells that are refractory to sorbitol uptake. Nevertheless, recordings at an increased gain managed to identify residual channel activity on some cells (*Figure 4H*). The currents on these cells corresponded to a 250- to 400-fold reduction in functional PSAC copy number, but the preserved inward-rectifying behavior suggests that the few remaining channels have unmodified unitary properties. As expected for cells with only a handful of active channels (*Conti and Wanke, 1975*), these ensemble traces exhibited increased channel noise. Remarkably, opening and closing transitions from individual channels were also detected despite the use of the whole-cell configuration (*Figure 3H*, traces beneath each ensemble).

We also used the cell-attached patch-clamp configuration to record single channels on the surface of infected erythrocytes and obtained quantitatively similar results (*Figure 4—figure supplement 1B*). Thus, both organic solute uptake and patch-clamp studies indicate that RhopH protein knockdown reduces PSAC copy number without altering the properties of the remaining channel molecules. Because knockdown of either protein affects the transfer of all three RhopH components to reinvaded erythrocytes, these *glmS* parasites could not determine the specific contributions of each component in channel formation; post-translational knockdown provided important insights, as described in a later section.

## Knockdowns validate the role of PSAC in nutrient uptake

Genetic mapping with a strain-specific PSAC inhibitor and with media containing altered nutrient concentrations has previously implicated a role for this channel in parasite nutrient acquisition (*Pillai et al., 2012*). We therefore measured parasite growth in our *glmS* knockdown parasites. We applied a nonsaturating 1 mM GlcN during gene transcription to allow direct comparison of parasite expansion rates when nutrient availability is altered. After invasion, matched numbers of ring-stage infected cells were collected and used for cultivation without GlcN in either standard RPMI 1640-based medium containing high nutrient concentrations or PGIM (*Pillai et al., 2012*); this modified medium follows the RPMI 1640 formulation but has reduced, more physiological concentrations of key nutrients that are acquired via PSAC. Our approach corrects for reduced *R3glmS* invasion and

quantifies the growth defect that results from reduced RhopH levels for a single parasite cycle. GlcN exposure significantly inhibited growth of *R2glmS* and *R3glmS* in standard media (*Figure 4I*, black bars; p<0.01 for comparisons to matched cultures without GlcN, *n* = 9 replicates from three independent trials each). Each transfectant's growth was compromised to a greater extent in the reduced-nutrient PGIM (red bars, p=0.02 when compared to growth reduction in standard medium). GlcN exposure did not affect wild-type parasites in either medium, excluding nonspecific toxicity. Greater growth inhibition in the nutrient-restricted PGIM parallels the enhanced potency of PSAC inhibitors in this medium and provides molecular validation of a role for PSAC in nutrient uptake at the host membrane.

## Knockdown as a result of protein destabilization defines contributions after invasion

Conditional knockdown using the *glmS* riboswitch is restricted to transcript-level regulation and, unfortunately, cannot distinguish the roles served by the encoded protein at distinct cell developmental stages or subcellular locations. This limitation proved to be especially problematic in our studies because both *glmS* knockdowns failed to transfer all RhopH components to new erythrocytes at invasion (*Figure 3B*), preventing the assignment of roles in PSAC formation to specific member proteins. To overcome this limitation and to achieve stage-specific knockdown, we tagged RhopH2 and RhopH3 with the *Escherichia coli* DHFR destabilization domain (DDD, *Supplementary file 1*). The DDD epitope conditionally interferes with target protein function as the fusion can be denatured by removal of trimethoprim (TMP), a stabilizing small molecule (*Iwamoto et al., 2010*); kinetic studies reveal that this approach permits post-translational knockdown in malaria parasites, but it is limited by a half-time of 6 hr for protein destabilization (*Muralidharan et al., 2011*). Repeated attempts to add DDD as an N-terminal tag after the predicted signal peptide cleavage sites were unsuccessful for both genes (*Supplementary file 1*), but integrations were readily obtained with C-terminal tagging (*Figure 5A*). With clones *R2-DDD* and *R3-DDD* carrying the C-terminal tag on RhopH2 and RhopH3 respectively, we observed stage-dependent effects on PSAC activity and parasite survival. Removal of TMP before the onset of gene transcription in the preceding intracellular cycle yielded a maximal reduction in channel activity ('max', *Figure 5B and C*, 87% ± 9% and 35% ± 2% decrease for *R2-DDD* and *R3-DDD*, respectively; p<0.001 for both parasites, n = 3–4 trials each).

Because RhopH2 knockdown yielded a greater loss of PSAC activity, we used *R2-DDD* to examine the effect of stage-specific TMP removal on protein trafficking and channel function. Using synchronized cultures, we removed TMP either after RhopH complex trafficking to rhoptries in late schizonts ('LS', *Figure 5B*) or immediately after erythrocyte invasion (early ring or 'ER'). A second stringent synchronization prior to TMP removal ensured that this ER culture did not undergo RhopH2 destabilization until after host cell invasion ('sync all', *Figure 5B*). Subsequent harvest at the mid-trophozoite stage revealed reduced permeability in both cultures (*Figure 5D and E*), with a greater reduction when TMP was removed before invasion (LS). As also seen with the *glmS* knockdowns, TMP removal before merozoite egress led to reduced delivery of CLAG3 and RhopH3 upon invasion (*Figure 5F*, LS), demonstrating that RhopH complex integrity is important for faithful transfer during invasion. Importantly, TMP removal after invasion preserved CLAG3 and RhopH3 levels (*Figure 5F*, ER). Nevertheless, these cells exhibited significantly reduced permeability (p=0.02, n=3), indicating that RhopH2 serves a role in channel formation beyond simply delivering the other RhopH components to invaded erythrocytes. The more modest effect on PSAC when TMP washout is performed after invasion may be conservatively accounted for by the relatively slow kinetics of DDD domain destabilization (*Muralidharan et al., 2011*).

We next used protease susceptibility studies to explore whether the ER-stage knockdown compromises CLAG3 delivery and insertion at the host membrane (*Figure 5G*). While 66% of CLAG3 in control cultures successfully inserted into the host membrane, TMP removal in early rings yielded reduced protease susceptibility (*Figure 5H*, p<0.01 from quantification of band intensities); TMP removal from late schizonts (LS) produced a similar reduction in CLAG3 insertion at the host membrane, compounding the reduced delivery during invasion described above. Thus, RhopH2 destabilization after invasion interferes with both CLAG3 membrane insertion and induction of PSAC activity.

The *R2-DDD* conditional knockdown provided additional molecular evidence for PSAC essentiality. Cultivation without TMP adversely affected the expansion of cultures within a single developmental cycle (red triangles, *Figure 5I*), indicating that any residual channel activity is insufficient to

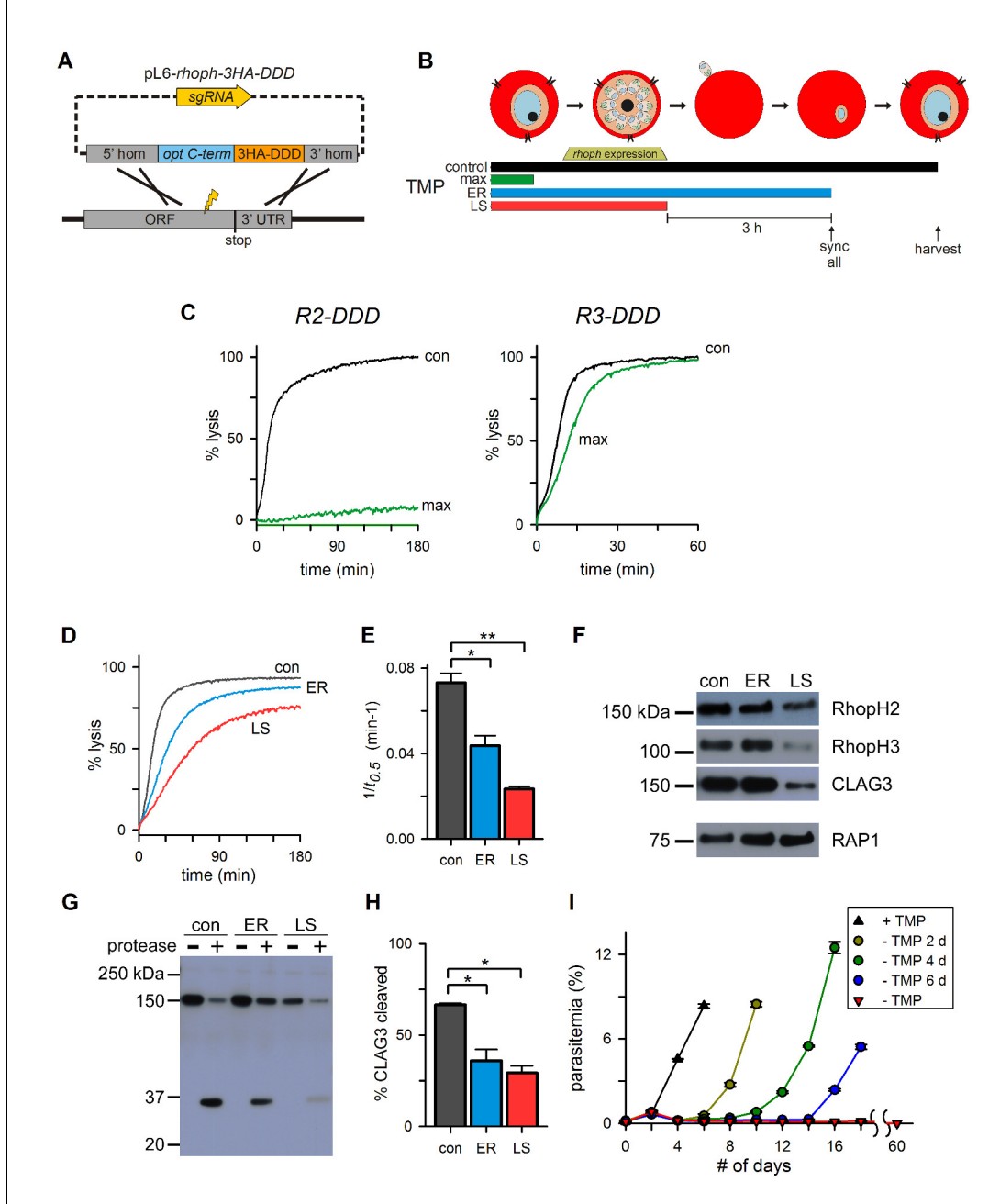

**Figure 5.** Post-translational knockdown reveals a requirement for RhopH2 after invasion. (**A**) Strategy for the introduction of a tandem 3xHA-DDD tag at the protein's C-terminus. (**B**) Schematic showing the timeline for the stage-specific removal of TMP and harvest of trophozoite-stage parasites. Horizontal bars below the schematic indicate the presence of TMP. control, TMP is present continuously; max, maximal knockdown is achieved with TMP removal prior to the onset of gene transcription; ER, TMP is removed after invasion and synchronization of early rings; LS, TMP is removed from late schizonts after protein trafficking to rhoptries. (**C**) Sorbitol-induced osmotic lysis kinetics for each transfectant after continuous cultivation with TMP (con) or withdrawal for maximal knockdown ('max', as illustrated in panel [B]). (**D**) Osmotic lysis kinetics for *R2-DDD* with TMP removal as indicated in panel (B). Traces are color-coded to match panel (B). (**E**) Mean ± S.E.M. of the reciprocal of sorbitol-induced osmotic lysis halftime ($t_{0.5}$) for control and each stage-specific *R2-DDD* knockdown. *p=0.02; **p<10$^{-3}$. (**F**) Immunoblots of trophozoite-stage *R2-DDD* parasites after stage-specific knockdown as indicated. While late-schizont TMP removal (LS) reduces each protein's abundance in newly invaded cells, early ring removal does not affect protein levels (ER). (**G**) Immunoblot showing CLAG3 protease susceptibility as an indicator of host membrane insertion for indicated stage-specific knockdowns. (**H**) Mean ± S.E.M. of fractional CLAG3 cleavage, quantified from three independent trials as in panel (G). *p<0.01. (**I**) Parasite expansion without or with TMP removal for two, four, or six days. Viable parasites could not be detected over a 60-day period when TMP was not restored.

meet parasite nutrient demand. Continued cultivation without TMP for two months did not produce outgrowth of resistant parasites, revealing that changes in other host- or parasite-derived nutrient uptake mechanisms cannot compensate for the loss of the RhopH-associated channel. Restoration of TMP after two, four, or six days permitted parasite recovery with increasing lag periods (circles), consistent with an increasing fraction of irreversibly affected parasites.

## Proteolytic processing of RhopH3 undermines conditional knockdown

We found that the relatively weak PSAC inhibition in *R3-DDD* upon TMP removal results from proteolytic processing of RhopH3 near its C-terminus (*Figure 6A*, arrow). Immunoblotting with an antibody against the 3xHA epitope tag added through transfection revealed both the unprocessed 130 kDa protein and a ~30 kDa soluble cleavage product (*Figure 6B*, red arrow). A RhopH3-specific antibody confirmed this processing as it detected two bands of 100 and 130 kDa, differing in size as predicted by the release of a single downstream fragment (*Figure 6A and C*). The downstream fragment's size implicates processing at 8–10 kDa upstream of the native C-terminus (*Figure 6A*, arrow). The fragment's presence only in schizonts and a relatively weak signal for the residual uncleaved form after invasion indicates near-complete processing soon after protein synthesis (*Figure 6B–D*). A strong anti-HA signal associated with rhoptries suggests processing at this site (*Figure 6D*), consistent with known protease activity in this organelle (*Sam-Yellowe et al., 2004*). The cleaved C-terminal fragment appears to be rapidly degraded, as it was absent from both spent culture medium and invaded cells (*Figure 6B*).

This proteolytic processing accounts for the modest PSAC knockdown observed in *R3-DDD* parasites. While TMP removal rendered all three RhopH proteins undetectable in the *R2-DDD* parasite, these proteins and their transfer to trophozoites were preserved in *R3-DDD* (*Figure 6E*). Immunoblotting revealed that the processed 100 kDa RhopH3 protein was minimally affected by cultivation without TMP for 48 hr; other members of the complex were also unaffected (*Figure 6F*). Probing with anti-HA revealed that the unprocessed 130 kDa RhopH3 isoform is TMP-sensitive, consistent with the idea that any RhopH3 that escapes proteolytic processing can be conditionally degraded through the DDD tag.

The biological significance of processing is unclear as uncleaved RhopH3 is still successfully transferred to the next erythrocyte upon invasion (*Figure 6B and D*, anti-HA signal). As imaging did not detect export of the unprocessed form from the PV (*Figure 6C*, anti-HA signal in trophozoites), one possibility is that processing facilitates RhopH3 export to the host cytoplasm.

## RhopH protein export to host cytosol is mediated by a protein translocon

Merozoite proteins that are transferred to the erythrocyte during invasion may be deposited onto the host membrane, directly into erythrocyte cytosol, or into the nascent PV surrounding the parasite (*Proellocks et al., 2010*). The RhopH complex appears to be secreted into the PV (*Figure 6D*), suggesting that its member proteins must cross the bounding PVM to enter host cytosol and eventually reach the erythrocyte surface. Most exported proteins cross this membrane via the *Plasmodium* translocon of exported proteins (PTEX), but a recent study reported that CLAG3 export is unaffected by conditional knockdown of HSP101 (*Beck et al., 2014*), an essential PTEX component. As PSAC activity was curtailed in the PTEX knockdown, other exported proteins may also be required to form the nutrient channel. Using the 13F10 HSP101-DDD clone (*Beck et al., 2014*), we confirmed that PTEX knockdown abolishes the development of PSAC-mediated uptake (*Figure 7A*) and we then examined RhopH protein export (*Figure 7B*). Remarkably, export of RhopH2 and of RhopH3 were both fully blocked in trophozoite-stage parasites grown for 48 hr without TMP. Colocalization with HRP2, a parasite protein carrying the canonical PEXEL export signal (*Beck et al., 2014*), showed that both proteins were trapped in the vacuole upon PTEX knockdown. In contrast to the earlier study, we also found that CLAG3 export requires PTEX activity as TMP removal prevented its movement out of the PV (*Figure 7B* bottom row; colocalization with SBP1, another PTEX substrate; *Beck et al., 2014*). Because these results depend on fixation and imaging conditions that may be aggravated by residual PTEX activity in the knockdown, we then used extracellular protease susceptibility to evaluate CLAG3 export. This revealed complete loss of CLAG3 protease susceptibility upon TMP removal (*Figure 7C*), sidestepping concerns about imaging conditions and indicating that

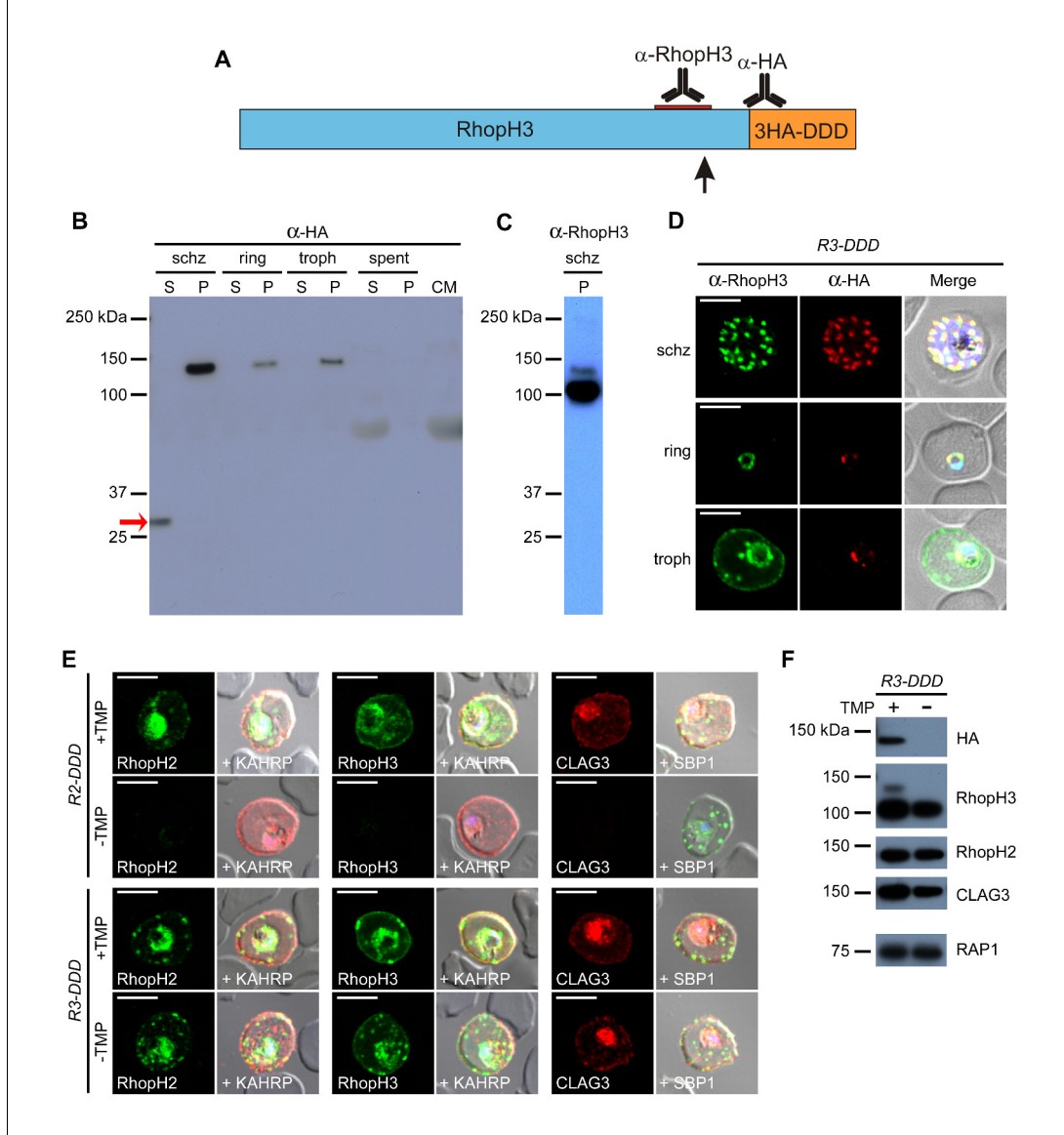

**Figure 6.** Proteolytic processing of RhopH3 compromises post-translational knockdown and accounts for the preserved PSAC activity in *R3-DDD* parasites. (**A**) Ribbon schematic showing RhopH3 (blue), the engineered C-terminal 3HA-DDD tag (orange), and the sites recognized by the indicated antibodies. The ribbon and the red bar indicating the epitope used to produce anti-RhopH3 antibody are drawn to scale. The data in this figure implicate proteolytic processing at a site marked with a black arrow. (**B**) Immunoblot of schizont-, ring-, and trophozoite-infected cell lysates from *R3-DDD* and of spent culture medium collected after invasion of synchronous cultures, probed with anti-HA. A red arrow points to the soluble cleavage product detected in schizonts. CM, culture medium not exposed to parasites. S and P, soluble and ultracentrifugation pellet fractions. (**C**) Immunoblot of a schizont-infected *R3-DDD* membrane fraction, probed with anti-RhopH3. A primary 100 kDa processed band and a less intense 130 kDa unprocessed band are apparent. (**D**) IFA of *R3-DDD* probed with the indicated antibodies at each parasite stage. Note that both antibodies detect protein in schizont rhoptries, but that only the anti-RhopH antibody detects export to the host membrane in trophozoites. Scale bars, 5 µm. (**E**) Trophozoite-stage IFAs of indicated parasites cultivated with and without TMP for 48 hr, showing that knockdown abolishes each member of the complex in *R2-DDD* but not in *R3-DDD*. Co-localization is shown with the exported parasite proteins KAHRP or SBP1 (red or green in merge panels, respectively). Scale bars, 5 µm. (**F**) Trophozoite-stage immunoblots using the indicated antibodies. *R3-DDD* was cultivated with or without TMP for 48 hr (the 'max' knockdown culture in *Figure 5B*).

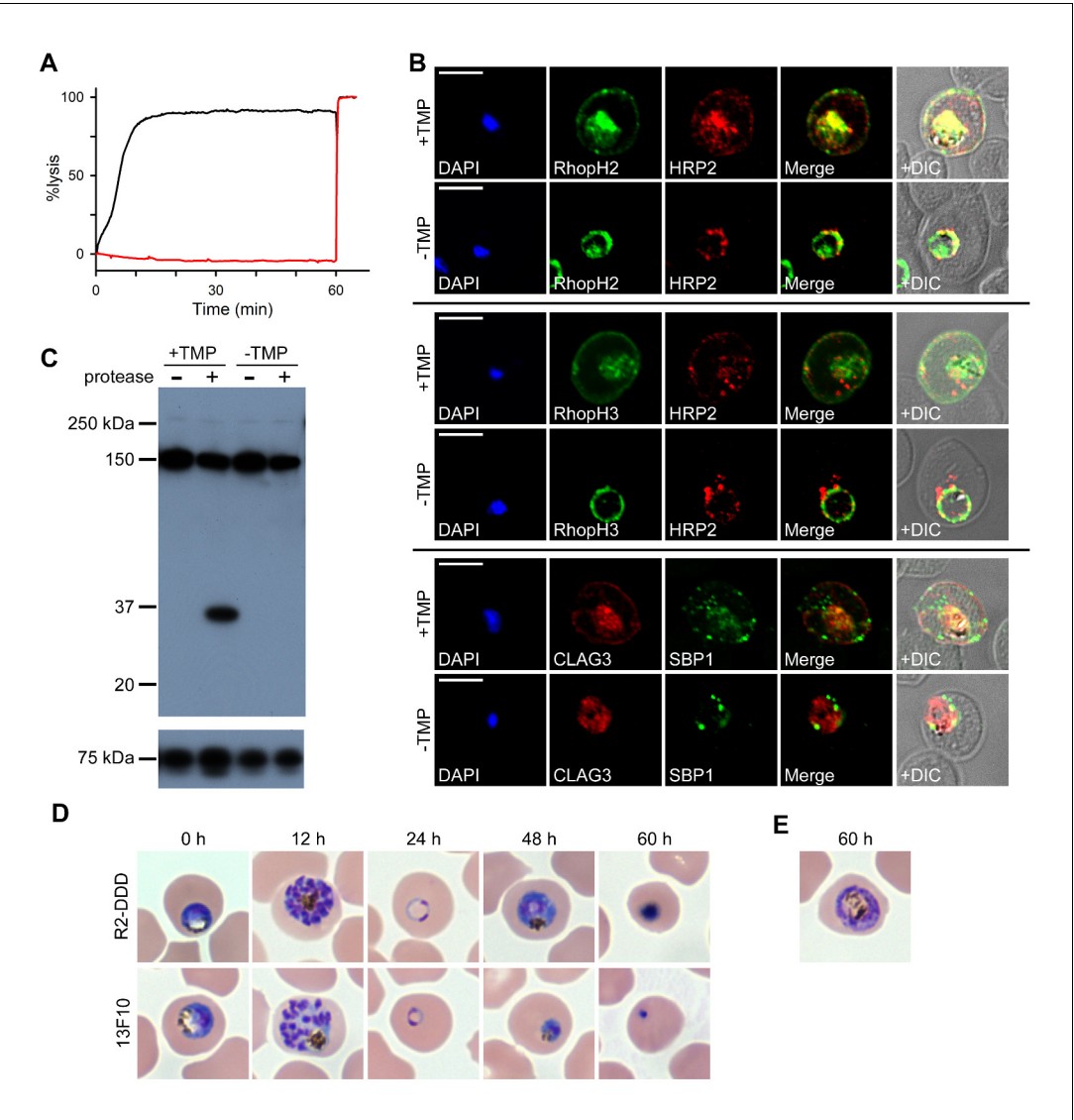

**Figure 7.** RhopH proteins are deposited in the PV and exported via PTEX. (A) Sorbitol-induced osmotic lysis of 13F10 parasites cultivated with or without 10 μM TMP (black and red traces, respectively). Cells were lysed by the addition of 0.5% saponin at 60 min. (B) IFA of 13F10 showing export of RhopH proteins after growth with or without TMP for 48 hr. Colocalization with exported HRP2 or SBP1 is indicated. Scale bars, 5 μm. (C) Immunoblot of 13F10 membrane fractions harvested after cultivation with or without TMP for 48 hr, probed with anti-CLAG3. A 35 kDa cleavage product of pronase E treatment indicates that CLAG3 is integral to the host membrane. (D) Giemsa-stained micrographs of *R2-DDD* and 13F10 at indicated intervals after TMP removal. (E) Alternate morphology of some arrested *R2-DDD* cells.

CLAG3 fails to reach the host membrane upon PTEX block. Thus, RhopH proteins are delivered to the PV upon invasion and then require PTEX for export at the PVM.

We next compared development in mid-trophozoite *R2-DDD* and 13F10 parasites subjected to TMP removal (*Figure 7D*). Both knockdown parasites completed their intracellular cycle and invaded new erythrocytes normally, consistent with already fulfilled requirements for protein export and nutrient uptake and with the relatively slow kinetics of DDD destabilization. After invasion, development was arrested in both parasites, with neither being capable of completing schizogony (*Figure 7D and E*). Interestingly, 13F10 parasites appeared to arrest at an earlier stage than *R2-DDD*

parasites (48 hr timepoint, *Figure 7D*). This earlier demise may reflect either contributions from the loss of other exported activities or a lower residual PSAC activity in 13F10 than in *R2-DDD* parasites.

## Discussion

We identify two distinct roles for RhopH proteins, which form a long-known but uncharacterized high-molecular weight complex in malaria parasites. As two effective sgRNAs targeting each gene could not produce knockouts, we employed both transcription- and protein-level conditional knockdown to dissect the trafficking requirements and stage-specific roles of the least well studied components, RhopH2 and RhopH3. Our studies implicate cotranslational assembly of a complex that also contains CLAG3. Subsequent trafficking of CLAG3 and RhopH2 to rhoptries depends on the formation of a complete and stable complex, but RhopH3 can reach this organelle in isolation. Upon merozoite egress, RhopH3 alone contributes to host-cell invasion, either by contributing to ligand interactions with erythrocyte receptors or by facilitating parasite internalization (*Figure 8*). Fully formed complexes, but not individual components, can then be transferred to the invaded erythrocyte and deposited into the nascent PV. These proteins then utilize PTEX to access erythrocyte cytosol. Subsequent trafficking via the Maurer's clefts, an organelle thought to facilitate onward protein export (*Tilley et al., 2008*), eventually inserts the complex into the host membrane. At the erythrocyte surface, this complex defines PSAC (*Figure 8*), a nutrient and ion channel that accounts for long-known increases in infected cell permeability (*Overman, 1948*; *Kutner et al., 1982*).

Our findings provide insights into the evolution of protein complexes in intracellular pathogens and account for the circuitous route taken by RhopH proteins to reach the host membrane. Early studies predicted that parasite proteins that are involved in the PSAC would be translated in ring- or trophozoite-stage parasites, just prior to the onset of solute uptake (*Kutner et al., 1982*). Genetic mapping identified CLAG3 as a PSAC determinant and contradicted these predictions (*Nguitragool et al., 2011*); production of CLAG3 in the prior cycle and packaging into rhoptries

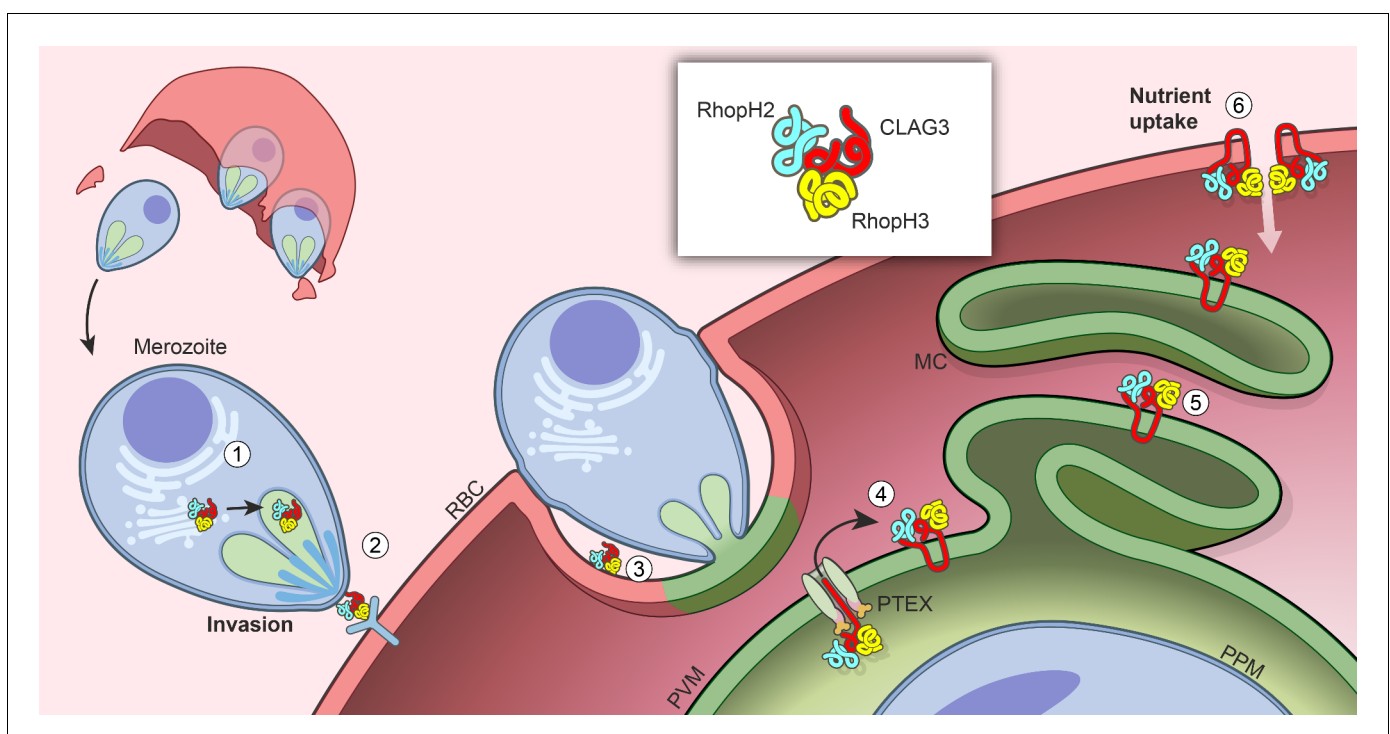

**Figure 8.** Model for trafficking of the RhopH complex and its roles throughout the parasite cycle. Cotranslational assembly of the three-member RhopH complex enables trafficking to the rhoptry ('1'). After merozoite egress, only RhopH3 contributes to invasion ('2'); the complex is then deposited into the PV ('3'), from where it is exported via PTEX ('4') and trafficked via the Maurer's clefts (MC, '5') to the host RBC membrane. There, the complex determines nutrient uptake via PSAC ('6').

seemed to be inconsistent with a primary role for CLAG3 in channel formation. Molecular evidence for a RhopH3 role in erythrocyte invasion, as we now provide, suggests that this unexpectedly early synthesis and complicated trafficking evolved to enable two distinct functions at separate points in the parasite cycle. Whether the role in invasion or in nutrient uptake arose first and how the second function was added is unclear as all three members of the RhopH complex are conserved in plasmodia but have no clear orthologs in other apicomplexan parasites. The RhopH complex presumably acquired both functions early in adaption to erythrocytes as both are essential for survival in the vertebrate bloodstream.

Preserved invasion upon RhopH2 knockdown along with the fortuitous combination of aborted CLAG3 but intact RhopH3 targeting to rhoptries in the *R2glmS* parasite allowed us to determine that RhopH3 is the only member of the complex involved in merozoite invasion. How might loss of this protein interfere with successful erythrocyte invasion? As *P. falciparum* utilizes multiple receptor-ligand interactions for invasion, one possibility is that RhopH3 interacts directly with an erythrocyte receptor. A yeast-two hybrid screen and binding studies with a recombinant C-terminal fragment of RhopH3 suggest that the anion exchanger Band3 protein might be the receptor (*Baldwin et al., 2014*), a proposal supported by the inhibitory effects of specific RhopH3 antibodies (*Ranjan et al., 2011*). Nevertheless, an indirect contribution of RhopH3 to ligand-receptor interactions or a role in transmembrane signaling to enable coordinated invasion remain good alternatives.

Our studies indicate that the RhopH complex is deposited into the PV upon invasion. It must then cross the PVM for trafficking onward to the host membrane. Immunofluorescence microscopy implicated the parasite-derived translocon termed PTEX in this process as all three RhopH complex members were blocked in the HSP101-DDD knockdown. Because our observations contradicted the original report involving this knockdown (*Beck et al., 2014*), we explored whether the timing of TMP withdrawal or subsequent cell harvest and fixation could account for this discrepancy, but found that a range of withdrawal times (late trophozoite- to mature schizont-stage cultures) and harvest times (24–36 hr after egress and invasion) produced indistinguishable results (not shown). Because we were unable to address this discrepancy, we also examined PTEX dependence by looking at CLAG3 protease susceptibility, an approach that revealed quantitative failure of CLAG3 delivery to the host membrane upon PTEX knockdown (*Figure 7C*).

Exactly how the RhopH complex interacts with PTEX remains unclear. Although these proteins lack the PEXEL motif present in most exported proteins, several other PEXEL-negative proteins also utilize PTEX to reach the erythrocyte cytosol (*Beck et al., 2014*; *Elsworth et al., 2016*). Do all three proteins require passage through the translocon? As subsequent trafficking and insertion at the host membrane depend on an intact RhopH complex (*Figure 5G*), models that require only a single RhopH protein to pass through the translocon may also yield the observed loss of export with PTEX knockdown. Is the complex disassembled by HSP101, a proposed unfoldase with an AAA+ ATPase domain, only to be reassembled after crossing the membrane? Because alkaline extraction suggests that RhopH proteins exist as both peripheral and integral membrane proteins within trophozoite-infected cells (*Figure 1A*), we speculate that passage through the translocon may also serve to fold and insert transmembrane domains into the bilayer. Indeed, most integral proteins in other organisms require translocons for faithful insertion into membranes; bacterial SecYEG and eukaryotic Sec61-type translocons achieve this by allowing correctly folded proteins to exit directly into the bilayer through a lateral gate (*Rapoport, 2007*). Whether this lateral translocation of integral proteins is possible in PTEX is unknown and requires study.

We propose that interaction with PTEX deposits RhopH2 and RhopH3 at the outer surface of the PVM (*Figure 8*). This orientation is consistent with accessibility studies that found that antibodies can bind RhopH2 on the PVM but not intra-vacuolar MSP1 without detergent permeabilization (*Hiller et al., 2003*). It is also consistent with subsequent export to the host membrane on exocytic vesicles, which predicts that proteins facing the host cytosol at the PVM will be endofacial when they reach the erythrocyte membrane. Immunoblotting confirms this final topology because RhopH2 and RhopH3 are not susceptible to extracellular protease (*Figure 1A*).

Quantitative loss of host cell permeability in RhopH2 and RhopH3 knockdowns addresses two important uncertainties about nutrient and ion uptake by infected cells. Whole-cell patch-clamp recordings on the conditional knockdowns reported here provide evidence against the induced upregulation of human channels by the parasite: near-complete removal of RhopH-associated channels showed stochastic transitions from the handful of remaining functional PSACs but failed to

uncover unrelated currents (*Figure 4H*; *Staines et al., 2007*). Non-conductive transport mechanisms such as transporters mediating electroneutral ion flux or those passing only uncharged solutes cannot be detected with patch-clamp and might still be present on infected cells; channels that are inactive under our recording conditions could also have been missed.

Although several studies have supported a CLAG3 role in PSAC formation (*Nguitragool et al., 2011*; *Pillai et al., 2012*; *Sharma et al., 2013*; *Mira-Martinez et al., 2013*), its paucity of predicted transmembrane domains and lack of homology to known channel proteins raised doubts about whether this protein could form a pore in isolation. Stable association with two other membrane proteins and their requirement for channel activity makes the direct formation of the channel pore by the RhopH complex more plausible. This model is also supported by genetic mapping of transport inhibition by proteases to the *clag3* locus (*Nguitragool et al., 2014*), by an amphipathic transmembrane domain that appears to line a water-filled pore (*Sharma et al., 2015*), and by site-directed mutagenesis of critical residues that define PSAC behavior (*Sharma et al., 2015*). Additional studies will be required to define the structural basis of permeation through PSAC and whether each RhopH member contributes to the pore or serves a less direct modulatory role.

Identification of a dual-function high-molecular weight complex with elaborate trafficking throughout the parasite cycle unveils a promising target for antimalarial therapies. PSAC activity is already an established drug target, validated with derivatives of a potent inhibitor and with molecular approaches (*Pillai et al., 2010*, *2012*); present studies). This channel is also the main exported activity cited as the basis of *in vitro* growth inhibition upon PTEX knockdown (*Beck et al., 2014*; *Elsworth et al., 2016*). The RhopH3 role in host cell invasion that we report provides additional motivation for drug or vaccine development. Inhibitors that interact with one or more members of this essential complex may disrupt disease progression by inhibiting both parasite invasion and nutrient acquisition.

## Materials and methods

### Parasite cultures
The genotype of *P. falciparum* KC5 clone (kindly provided by Thomas Wellems) was confirmed by DNA sequencing. This line and engineered derivative clones were grown in $O^+$ human erythrocytes (Interstate Blood Bank, Inc.) by standard methods and maintained at 5% hematocrit under 5% $O_2$, 5% $CO_2$, 90% $N_2$ at 37°C.

### Plasmid design and construction
Plasmids were designed and constructed to allow CRISPR-Cas9 transfection of cultivated parasites to produce *rhoph*-positive control integrants, knockouts, and conditional knockdowns carrying either a *glmS* ribozyme in the 3'UTR or a tandem 3xHA-DDD encoded after the predicted signal peptide cleavage site or as a C-terminal tag. The pL6 plasmid was modified with In-Fusion cloning (Clontech) to achieve sgRNA expression under the U6 spliceosomal RNA promoter as described (*Ghorbal et al., 2014*). Genomic sequences for homologous recombination repair were obtained as synthetic DNA constructs and inserted into the pL6 plasmid using In-Fusion cloning. Shield mutations were introduced on the donor sequence at sites corresponding to the genomic sgRNA target by the use of codon-optimized synthetic DNA; these mutations prevent continued cleavage by the sgRNA-Cas9 complex and provide purifying selection for desired integrant parasites. For gene knockouts, the donor sequence homology arms flanked an in-frame full-length human dihydrofolate reductase (*hDHFR*) gene to allow selection of possible integrants with WR99210; for positive control transfections, a codon-optimized sequence that does not alter the native protein sequence was used in place of the *hDHFR* gene and flanked by identical homology arms. Cas9 was expressed from an unmodified pUF1-Cas9 plasmid carrying the yeast dihydroorotate dehydrogenase (*yDHODH*) cassette for selection with DSM1 (*Ghorbal et al., 2014*). Primers used are listed in *Supplementary file 1*. DNA sequencing and restriction digestion were used to confirm all constructed plasmids.

### *P. falciparum* strain generation
Transfections were initiated by electroporation of pUF1-Cas9 and modified pL6 plasmids into uninfected erythrocytes prior to use for parasite cultivation. The transfected culture was selected with

1.5 µM DSM1; in some cases, selection with 1 nM WR99210 was used to ensure retention of the pL6 and to favor the growth of possible knockout parasites expressing *hDHFR* from genomic sites. After 2–3 weeks, parasite growth was detected by Giemsa-staining. PCR was used to evaluate the integration of the donor sequence from the pL6 plasmid, the retention of wild-type genomic sequence, and the presence of pL6 episomes. All experiments were performed with limiting dilution clones (*Lyko et al., 2012*).

Conditional knockdown of RhopH proteins in the *R2glmS* and *R3glmS* parasites was achieved by addition of GlcN to synchronous trophozoite-stage cultures (*Prommana et al., 2013*). GlcN exposure was continued for up to 24 hr before washout and continued cultivation until harvest for phenotype studies. Control experiments revealed that GlcN addition under these conditions does not produce measurable toxicity in wild-type parasites at up to a 4 mM concentration (not shown).

*R2-DDD* and *R3-DDD* parasite clones were continuously propagated with 10–20 µM TMP, a concentration that is nontoxic to untransfected KC5 parasites as its PfDHFR carries trimethoprim-resistance mutations. Conditional knockdown was initiated by TMP removal at indicated timepoints in the parasite cycle.

The 13F10 HSP101-DDD clone was cultivated continuously with 10 µM TMP and 2.5 µg/mL blasticidin S as described (*Beck et al., 2014*). Knockdown was initiated by TMP removal at the trophozoite stage; these parasites were cultivated for 48 hr prior to harvest for osmotic lysis assays, indirect immunofluorescence assays, or immunoblotting. PCR was used to confirm the integrity of the single homologous recombination cassette prior to all phenotype studies.

## Antibody production

Rabbit polyclonal antibodies targeting RhopH2 and RhopH3 were produced using Genomic Antibody Technology (SDIX). Rabbits were immunized with DNA plasmids encoding fragments of RhopH2 (residues I1159 to F1308) or RhopH3 (residues T731 to Y829) in an OLAW and AAALAC accredited facility. The immune response was boosted with recombinant proteins expressed in *E. coli.* Antibodies were affinity-purified on cognate antigens and verified by ELISA and immunoblotting.

## Immunofluorescence microscopy

Indirect immunofluorescence assays were performed using air-dried thin smears prepared on microscope slides and fixed with 50:50 acetone-methanol. After fixation, slides were blocked with 3% skim milk powder in PBS for 30 min before incubation with primary antibodies in blocking buffer (mouse anti-CLAG3, 1:100; rabbit anti-RhopH2, 1:200; rabbit anti-RhopH3, 1:500; mouse anti-HRP2, 1:500; mouse anti-KAHRP, 1:100; rabbit anti-SBP1, 1:500; rabbit anti-RAP1, 1:3000; mouse anti-RAP1, 1:100; mouse anti-HA, 1:100) for 1 hr at RT. RAP1 antibodies have been described previously (*Ito et al., 2011*). Primary antibodies were detected with Alexa Fluor 488 or 594 secondary IgG antibodies (Thermo Fisher Scientific, Waltham, MA) at a 1:500 dilution. Images were collected on a Leica SP5 microscope under a 64x oil immersion objective with serial 405 nm, 488 nm, or 594 nm excitations. Images were processed using Leica LAS X software.

## Immunoblotting

Immunoblots were obtained using cultivated parasites, with or without pronase E treatment as described previously (*Nguitragool et al., 2011*). Synchronous parasite cultures were used either as total cell lysates or after membrane fractionation using hemolysis in lysis buffer (7.5 mM $Na_2HPO_4$, 1 mM EDTA, 1 mM PMSF, pH 7.5) and ultracentrifugation (100,000 x g, 4°C, 1 hr). The supernatant was considered the 'soluble' fraction. Where used, the membrane pellet was subjected to carbonate extraction by resuspension in 100 mM $Na_2CO_3$, pH 11 at 4°C for 30 min before a second ultracentrifugation to separate peripheral from integral membrane proteins. Samples were solubilized and reduced in a modified Laemmli sample buffer containing a final 6% SDS concentration. Proteins were separated by electrophoresis in a 4–15% Mini-PROTEAN TGX gel (Bio-RAD) and transferred to nitrocellulose membranes. After blocking (using 3% skim milk powder in 150 mM NaCl, 20 mM TrisHCl, pH 7.4 with 0.1% Tween20), primary antibodies were applied at 1:1000 to 1:3000 dilution in blocking buffer. After washing, HRP-conjugated secondary antibodies were applied at 1:3000 dilution with chemiluminescent substrate (Clarity Western ECL substrate, Bio-Rad).

## Electrophysiology

Patch-clamp recordings from the infected erythrocyte membrane and the PVM were obtained using low-noise quartz capillaries pulled to pipette resistances of 1–3 MΩ in symmetric bath and pipette solutions of 1000 mM choline chloride, 115 mM NaCl, 10 mM MgCl$_2$, 5 mM CaCl$_2$, 20 mM Na-HEPES, pH 7.4 as described previously (*Desai et al., 2000*). Freshly prepared patch-clamp pipettes typically yielded seal resistances >100 GΩ on infected erythrocytes. Single channel recordings were obtained in the cell-attached configuration with a holding potential of 0 mV. Indicated membrane potentials were imposed as steps from this holding potential using an Axopatch 200B amplifier and a Digidata 1550 digitizer (Molecular Devices). Whole-cell patch-clamp recordings were obtained after application of brief, high voltage electrical pulses to disrupt the membrane patch beneath the pipette.

Patch-clamp of the parasitophorous vacuolar membrane (PVM) used identical pipettes and recording solutions. This membrane was accessed by use of mechanical suction applied to the pipette after obtaining a low resistance seal on the erythrocyte membrane as described (*Desai et al., 1993*).

All acquired data were low-pass filtered at 5 kHz (8-pole Bessel, Frequency Devices), digitized at 100 kHz, and recorded with Clampex 10.0 software (Molecular Devices). Recordings were analyzed with Clampfit 10.0 software (Molecular Devices) or locally developed code.

## Parasite growth inhibition and invasion assays

Parasite growth was quantified using SYBR Green I fluorescence and is based on the detection of parasite nucleic acids, as described previously (*Pillai et al., 2012*). After conditional knockdown of RhopH protein in trophozoite-stage parasites with 1 mM GlcN treatment for 24 hr (*R2glmS*, *R3glmS*, and control untransfected wild-type), cultures were sorbitol synchronized to ensure seeding of ring-stage parasites at 0.5% parasitemia and 2.5% hematocrit in 96 well plates. These cultures were maintained for 72 hr at 37°C in standard medium or PGIM without GlcN or medium changes. They were then lysed in buffer containing 20 mM Tris, 10 mM EDTA, 0.016% saponin, 1.6% Triton X-100, pH 7.5 with SYBR Green I nucleic acid stain (Invitrogen) at a 5000-fold dilution. After a 30 min incubation in the dark, DNA was quantified with fluorescence measurements (excitation and emission at 485 and 528 nm, respectively). Matched controls without GlcN and with 20 µM chloroquine were used as positive and negative controls, respectively, for the normalization of growth replicates in each experiment.

Merozoite invasion in *glmS* knockdown parasites was evaluated using tightly synchronized cultures after treatment with indicated GlcN concentrations beginning at the trophozoite stage. After 24 hr, triplicate Giemsa-stained smears were examined for retained schizonts, extracellular merozoites, and ring-stage parasites as a marker of successful invasion.

## Osmotic lysis assays

Organic solute uptake by infected cells was continuously tracked as described previously (*Nguitragool et al., 2011*). Trophozoite-stage infected cells were enriched with the percoll-sorbitol method, washed, and resuspended in 150 mM NaCl, 20 mM Na-HEPES buffer, 0.1 mg/mL BSA, pH 7.4. Solute uptake was initiated by the addition of an osmotic lysis solution containing 280 mM sorbitol, 20 mM Na-HEPES, 0.1 mg/mL BSA, pH 7.4. Permeability studies for other solutes utilized iso-osmolar replacement of sorbitol in this solution. The kinetics of osmotic lysis were monitored by recording the transmittance of 700 nm light through the cell suspension. Addition of 0.5% saponin at the end of each recording was used for normalization of transmittance values to 100% cell lysis. Osmotic lysis half-times, normalized permeabilities, and percentage of cells refractory to osmotic lysis were determined with locally developed code.

Knockdown parasites were percoll-enriched after preincubation in percoll-sorbitol or percoll-xylitol solutions to allow efficient recovery of cells that had reduced organic solute permeabilities as described previously (*Hill et al., 2007*).

## Statistical analysis

All numerical data were calculated and plotted as mean ± S.E.M. Statistical significance was calculated by unpaired Student's t-test or one-way ANOVA. Significance was accepted at $p < 0.05$ or indicated values.

## Acknowledgements

We thank Josh Beck, Daniel Goldberg, Thomas Wellems, Jose-Juan Lopez-Rubio, Philip Shaw, and Takafumi Tsuboi for providing plasmids, antibodies, and/or parasite clones; Josh Beck and Daniel Goldberg for helpful discussions and sharing their unpublished results on PTEX activity; Nicole Potchen and Meera Garriga for critical reading of the manuscript; and Ryan Kissinger and Anita Mora for artwork. DSM1 (MRA-1161) was obtained through MR4 as part of the BEI Resources Repository, NIAID, NIH. SAD is a named inventor on US and international patent applications and on an issued US patent on the drug targets presented in this manuscript ('INHIBITORS OF THE PLASMODIAL SURFACE ANION CHANNEL AS ANTIMALARIALS', International Patent Application Publication WO/2010/011537; Chinese Patent Application 200980137435; European Patent Application EP2313100A1; Indian Patent Application 470/CHENP/2011; US Patent Application Publication US2014/0088082A1; and issued US Patent US8,618,090). The other authors have no conflicts of interest. This research was supported by the Intramural Research Program of the National Institutes of Health, National Institute of Allergy and Infectious Diseases.

## Additional information

### Competing interests

SAD: Is a named inventor on US and international patent applications and an issued US patent on the drug targets presented in this manuscript ("Inhibitors of the plasmodial surface anion channel as antimalarials", International Patent Application Publication WO/2010/011537; Chinese Patent Application 200980137435; European Patent Application EP2313100A1; Indian Patent Application 470/CHENP/2011; US Patent Application Publication US2014/0088082A1; and issued US Patent US8,618,090). The other authors declare that no competing interests exist.

### Funding

| Funder | Grant reference number | Author |
| --- | --- | --- |
| Division of Intramural Research, National Institute of Allergy and Infectious Diseases | Intramural Research Support | Daisuke Ito<br>Marc A Schureck<br>Sanjay A Desai |

The funders had no role in study design, data collection and interpretation, or the decision to submit the work for publication.

### Author contributions

DI, Conceptualization, Data curation, Formal analysis, Investigation, Writing—original draft, Writing—review and editing; MAS, Data curation, Formal analysis, Investigation, Writing—original draft, Writing—review and editing; SAD, Conceptualization, Data curation, Formal analysis, Supervision, Writing—original draft, Writing—review and editing

### Author ORCIDs

Marc A Schureck, http://orcid.org/0000-0001-9880-0783
Sanjay A Desai, http://orcid.org/0000-0003-2150-2483

### Ethics

Animal experimentation: Polyclonal antibody production in rabbits was performed in strict accordance with the recommendations in the Guide for the Care and Use of Laboratory Animals of the National Institutes of Health by SDIX, a facility accredited by OLAW (Assurance #: A3975-01) and AAALAC (Delaware Accreditation # 00806).

## Additional files

**Supplementary files**
• Supplementary file 1. Primers used in this study. Underlined sequences represent overhangs for In-Fusion cloning (Clontech).

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
