## [Decision Letter]

Thank you for submitting your article "An essential dual-function complex mediates both host invasion and nutrient uptake in malaria" for consideration by *eLife*. Your article has been reviewed by two peer reviewers, one of whom, Jon Clardy (Reviewer #1), is a member of our Board of Reviewing Editors, and the evaluation has been overseen by Richard Aldrich as the Senior Editor.

The reviewers have discussed the reviews with one another and the Reviewing Editor has drafted this decision to help you prepare a revised submission.

The reviewers were uncertain about the suitability of this manuscript for *eLife*, and after some discussion agreed on revision. A revision of your manuscript, if you should elect to pursue that path, will need to successfully address two general issues that pervade the current version of the manuscript. The first is the writing in the current version, which both reviewers found difficult to understand throughout and verging on the impenetrable on occasion. Much of this lack of clarity comes from trying to provide a comprehensive list of a lot of experiments without regard to providing a clearly discernible story line. It comes across as poorly organized, full of puzzling figures, and the occasional poorly supported claim. There is no clear and systematic definition of phenotypes and that coupled with different reverse genetic approaches contribute to the lack of clarity. Your report needs to be more focused, and you need to clearly define the contribution of the two components you focus on to the two phenotypes, invasion and nutrient uptake, involved.

Second, you need to make the article more accessible and interesting to researchers that are not familiar with (and possibly not terribly interested in) malaria. Unsupported, or unlikely statements, that your findings will contribute to therapeutic interventions, or provide a "framework for the evolution of protein complexes", or provides "direct molecular evidence for a RhopH complex role in parasite invasion" need to be fully justified or removed.

*Reviewer #1:*

Summary: In this manuscript the authors characterize the roles of RhopH2 and RhopH3, two of the three proteins in the *Plasmodium falciparum* RhopH complex. They, and others, had previously demonstrated that CLAG3 (aka RhopH1) was important for solute uptake, and they tested if RhopH2 and RhopH3 also played a role in solute uptake. Additionally, other reports had implicated the RhopH complex (and in particular RhopH3) in the erythrocyte invasion, so they assessed the importance of RhopH3 and possibly of RhopH2 for invasion.

In accordance with previous studies (Baldwin 2014 and Ranjan 2011), they found that Rhop3 (and Rhop2) was important for invasion, although the studies were much more definitive for Rhop3 than Rhop2. Their studies, while not surprising, do provide important supporting information for the generally accepted model.

Their finding that both Rhop2 and Rhop3 are essential for the successful transfer of each of the RhoP proteins into a new erythrocyte host is a new finding. Their finding that Rhop2 helps form functional plasmodial surface anion channels (PSACs) is also new, although the effect of Rhop2 degradation after invasion was modest. They also report the paradoxical – in the sense that it contradicts earlier reports – result that the RhopH complex uses the PTEX transporter to leave the parasitophorous vacuole (PV). Their discussion clearly addresses but does not resolve the contradiction.

Issues to be addressed in a revision

1) The authors refer repeatedly to their use of CRISPR technology to define the roles of RhopH2 and Rphoh3, and they do demonstrate CRISPR's utility in *P. falciparum*. But CRISPR's main utility was to introduce older technologies by creating the transcription knockdowns and conditional protein destabilization domains that provided most of the significant results in this manuscript. And CRISPR's use in Plasmodium has already been well documented (Ghorbal 2014).

2) The authors stress that their findings "suggest several novel therapeutic approaches", and they offer one in "drugs that interfere with assembly of the RhopH complex." This approach seems to require finding protein-protein interaction inhibitors (PPIs), which is a field with few successes and numerous failures. I believe that they should temper their claims, or possibly cite what success they've had in the approach as they do have a patent application.

3) The article is very difficult to read as it is written for an audience of experienced malaria researchers. The manuscript desperately needs a thorough editing to make it more accessible to *eLife* readers. For example, the term 'rhoptry' is used both explicitly and implicitly throughout the article – "rhop" appears more than 100 times in the main text – and I'm certain that few *eLife* readers know what a rhoptry is.

4) The authors explain that RhopH3 proteolytic processing (Figure 6) is responsible for the lack of lysis loss observed in the R3-DDD case (Figure 4). This explanation implies that RhopH3 is not being degraded by the DDD domain, and therefore active RhopH3 remains. However, the authors do not show that active RhopH3 is abundant in R3-DDD without TMP. I believe this should be an easy experiment to perform (the same style of blots they performed for the R2-DDD case should be sufficient). And this experiment would serve as great confirmation of their model that the minimal affect of R3-DDD is due simply to the method not working well to degrade RhopH3. If instead they observe substantial degradation of RhopH3, then the DDD domain likely degrades RhopH3 to a significant degree before the proteolytic processing removes the DDD domain. Then the authors would need to propose another explanation for why the removal of RhopH3 has minimal impact on PSAC function.

Less important issues

5) In the Abstract, the authors claim that their conditional knockdowns demonstrate that the rhoph2 and rhoph3 genes are required for bloodstream survival. Should this not read "in vitro" survival? There were no experiments within blood, so this seems like an overstatement.

6) The authors should reference Comeaux et al., Mol Microbiol 2011, 80:378. This paper shows that a knockout of CLAG3 had growth defects, which is an important corroboration of the Cell 2011 paper.

7) In subsection “Assembly-dependent trafficking and a RhopH3 role in invasion” at the end of third paragraph, the statement "Thus, all three RhopH complex members require interactions with their cognate partners for successful transfer to erythrocytes at invasion." should be changed to make it clear that the data do not suggest that RhopH2 and RhopH3 require CLAG3 in order to get transferred. That is apparently not necessary, as CLAG3 knockouts are viable (Comeaux reference above).

8) The submission letter states that single molecule studies have been performed, but I can't figure out what this refers to.

*Reviewer #2:*

The paper by Ito et al. describes functional analysis of the 2 essential proteins RhopH2 and RhopH3 in the malaria parasite *Plasmodium falciparum*. For this purpose the authors use two different conditional knock-down approaches and identify a role in parasite invasion and nutrient uptake.

This is a very interesting study and following from their previous work on the parasite-induced channel PSAC. Data interpretation is complicated because both knock-downs also affect expression, and therefore likely function, of each other and a third PSAC component CLAG3. This should be accounted for more carefully for data interpretation and stated more explicitly throughout the manuscript.

Figure 2/D: It is very confusing to see that RhopH2 and RhopH3 knock-down results in simultaneous complete loss of each other's expression and CLAG3, but this is not recapitulated by IFA: here only CLAG3 is co-depleted in the knock-downs. It is also not clear whether RhopH2 co-localization with Rap1 is affected in the RhopH3 knock-down but not vice versa, as stated. This conflicting data and subsequent interpretation is a major issue continuing throughout the entire MS that needs to be thoroughly addressed.

Figure 3: This figure again shows the possible combinatorial function and cumulative phenotypes of the 2 knock-downs vs the 3 affected proteins. It should be a manor goal of the revision to address this issue and untangle the phenotypes/functions.

---

## [Author Response]

*Reviewer #1:*

*Summary: In this manuscript the authors characterize the roles of RhopH2 and RhopH3, two of the three proteins in the Plasmodium falciparum RhopH complex. They, and others, had previously demonstrated that CLAG3 (aka RhopH1) was important for solute uptake, and they tested if RhopH2 and RhopH3 also played a role in solute uptake. Additionally, other reports had implicated the RhopH complex (and in particular RhopH3) in the erythrocyte invasion, so they assessed the importance of RhopH3 and possibly of RhopH2 for invasion.*

*In accordance with previous studies (Baldwin 2014 and Ranjan 2011), they found that Rhop3 (and Rhop2) was important for invasion, although the studies were much more definitive for Rhop3 than Rhop2. Their studies, while not surprising, do provide important supporting information for the generally accepted model.*

We do not agree that RhopH protein involvement in invasion is generally accepted. These and other prior studies have indirectly supported invasion roles for these proteins as well as CLAG3, relying primarily on erythrocyte binding assays with recombinant peptides and growth inhibition assays using antibodies. Unfortunately, these assays often produce misleading results because of nonspecific peptide binding and well-documented studies showing antibody cross-reactivity for parasite antigens (Holmquist et al., 1988; Udomsangpetch et al., 1989). Notably, while studies have suggested CLAG3 and RhopH2 contribute to invasion (Sam-Yellowe et al., 1992; Ocampo et al., 2005), our molecular studies formally refute this presumed role for both of these proteins. The revised manuscript clarifies this issue with a better description of prior data and highlights the new insights provided by our study (Results section).

We have performed additional experiments and revised the Results to clarify that only RhopH3 contributes to invasion (Figure 2 and subsection “Knockdowns co-deplete CLAG3 and reveal that only RhopH3 contributes to invasion”). We now show that RhopH3 knockdown does not affect merozoite development or parasite egress, and that its role is restricted to either host cell interaction or subsequent parasite internalization.

*Their finding that both Rhop2 and Rhop3 are essential for the successful transfer of each of the RhoP proteins into a new erythrocyte host is a new finding. Their finding that Rhop2 helps form functional plasmodial surface anion channels (PSACs) is also new, although the effect of Rhop2 degradation after invasion was modest. They also report the paradoxical – in the sense that it contradicts earlier reports – result that the RhopH complex uses the PTEX transporter to leave the parasitophorous vacuole (PV). Their discussion clearly addresses but does not resolve the contradiction.*

We agree with this assessment. The effect of RhopH2 degradation after invasion was significant (P = 0.02, Student’s t test with n = 3 independent, matched trials), and modest only when compared to the nearly complete loss of PSAC seen with protein degradation prior to invasion. This lesser magnitude probably reflects the slow kinetics of protein destabilization/degradation after trimethoprim removal (halftime of 6 h for soluble proteins (Muralidharan et al., 2011), presumably longer for membrane proteins such a RhopH2). We have revised subsection “Knockdown via protein destabilization defines contributions after invasion” to clarify this point.

We heartily thank Josh Beck and Daniel Goldberg for multiple discussions and free exchange of reagents, both critical for examining the discrepancy with the prior report on PTEX. We agree that this contradiction will require further study.

*Issues to be addressed in a revision*

*1) The authors refer repeatedly to their use of CRISPR technology to define the roles of RhopH2 and Rphoh3, and they do demonstrate CRISPR's utility in P. falciparum. But CRISPR's main utility was to introduce older technologies by creating the transcription knockdowns and conditional protein destabilization domains that provided most of the significant results in this manuscript. And CRISPR's use in Plasmodium has already been well documented (Ghorbal 2014).*

We have revised the text to minimize references to CRISPR. We agree that this technology is already documented in malaria research, but to our knowledge, our study represents the first to perform a large number of CRISPR transfections to define and dissect the roles of two essential, interacting proteins.

*2) The authors stress that their findings "suggest several novel therapeutic approaches", and they offer one in "drugs that interfere with assembly of the RhopH complex." This approach seems to require finding protein-protein interaction inhibitors (PPIs), which is a field with few successes and numerous failures. I believe that they should temper their claims, or possibly cite what success they've had in the approach as they do have a patent application.*

We have tempered these claims.

*3) The article is very difficult to read as it is written for an audience of experienced malaria researchers. The manuscript desperately needs a thorough editing to make it more accessible to eLife readers. For example, the term 'rhoptry' is used both explicitly and implicitly throughout the article – "rhop" appears more than 100 times in the main text – and I'm certain that few eLife readers know what a rhoptry is.*

We have edited the manuscript to make more accessible to the broad *eLife* readership. The Introduction now also includes a more complete overview of the relevant background on parasite cell biology, a description of the key unknowns that we resolved, and a description of methods used. We have also added new schematics to help our readers (Figure 2, Figure 3, Figure 4, Figure 5, Figure 6, and Figure 4—figure supplement 1).

*4) The authors explain that RhopH3 proteolytic processing (Figure 6) is responsible for the lack of lysis loss observed in the R3-DDD case (Figure 4). This explanation implies that RhopH3 is not being degraded by the DDD domain, and therefore active RhopH3 remains. However, the authors do not show that active RhopH3 is abundant in R3-DDD without TMP. I believe this should be an easy experiment to perform (the same style of blots they performed for the R2-DDD case should be sufficient). And this experiment would serve as great confirmation of their model that the minimal affect of R3-DDD is due simply to the method not working well to degrade RhopH3. If instead they observe substantial degradation of RhopH3, then the DDD domain likely degrades RhopH3 to a significant degree before the proteolytic processing removes the DDD domain. Then the authors would need to propose another explanation for why the removal of RhopH3 has minimal impact on PSAC function.*

This is an important suggestion. We have now performed the requested immunoblots (Figure 6). We also performed new IFAs showing that each RhopH member is abolished in R2-DDD trophozoites, but preserved in R3-DDD (Figure 6). These findings now show that proteolytic processing accounts for the relatively modest loss of PSAC activity in the R3-DDD knockdown.

We added a new schematic summarizing these findings (Figure 6) and revised the text accordingly (subsection “Proteolytic processing of RhopH3 undermines conditional knockdown”).

Less important issues

*5) In the Abstract, the authors claim that their conditional knockdowns demonstrate that the rhoph2 and rhoph3 genes are required for bloodstream survival. Should this not read "*in vitro*" survival? There were no experiments within blood, so this seems like an overstatement.*

We agree and have changed this.

*6) The authors should reference Comeaux et al., Mol Microbiol 2011, 80:378. This paper shows that a knockout of CLAG3 had growth defects, which is an important corroboration of the Cell 2011 paper.*

We have added this citation.

*7) In subsection “Assembly-dependent trafficking and a RhopH3 role in invasion” at the end of third paragraph, the statement "Thus, all three RhopH complex members require interactions with their cognate partners for successful transfer to erythrocytes at invasion." should be changed to make it clear that the data do not suggest that RhopH2 and RhopH3 require CLAG3 in order to get transferred. That is apparently not necessary, as CLAG3 knockouts are viable (Comeaux reference above).*

The reported CLAG3 knockout and a tandem CLAG3-CLAG2 knockdown we reported (Hill 2007, Sharma 2013) retain paralogs on other chromosomes (CLAG8 and CLAG9 at a minimum). Because RhopH2 and/or RhopH3 transfer may be facilitated by interactions with any remaining CLAG paralog, it is not currently possible to exclude a similar CLAG requirement as the reviewer suggests. The revised manuscript clarifies this point.

*8) The submission letter states that single molecule studies have been performed, but I can't figure out what this refers to.*

We apologize for this vague terminology. We were referring to single channel recordings with patch-clamp. We show currents from individual channels on the host erythrocyte membrane (Figure 4—figure supplement 1) and on the parasitophorous vacuolar membrane surrounding the intracellular parasite (Figure 3). We have revised the text (subsections “Activities at the parasitophorous vacuole are unaffected” and” The RhopH complex is required for nutrient channels on the host membrane”), Figure 3, Figure 4, and the Figure 3—figure supplement 1 and Figure 4—figure supplement 1 to clarify these studies and the insights they provide.

*Reviewer #2:*

*[…] Figure 2/D: It is very confusing to see that RhopH2 and RhopH3 knock-down results in simultaneous complete loss of each other's expression and CLAG3, but this is not recapitulated by IFA: here only CLAG3 is co-depleted in the knock-downs. It is also not clear whether RhopH2 co-localization with Rap1 is affected in the RhopH3 knock-down but not vice versa, as stated. This conflicting data and subsequent interpretation is a major issue continuing throughout the entire MS that needs to be thoroughly addressed.*

We apologize for the confusing presentation, which resulted from an inadequate explanation of the timing of key biological events. We have overhauled the Results section, added new data, and modified these figures to clarify the complicated interactions between these proteins and the stage-specific effects of knockdown.

We found that transcriptional knockdown of either protein has both early effects on trafficking of associated proteins within merozoites AND late effects on transfer of the complex to the next erythrocyte. These early and late effects are now presented separately in revised Figure 2 and Figure 3, respectively.

The early effects were measured within 12 h of glucosamine addition when intracellular parasites have matured to schizonts but prior to egress. At this stage, our studies show that RhopH2 knockdown renders CLAG3 undetectable in both IFA and immunoblots, but RhopH3 is largely preserved and traffics normally to apical organelles in developing merozoites (revised Figure 2, where we have added a new stage-specific immunoblot). RhopH3 knockdown also renders CLAG3 undetectable (Figure 2), but differs in that RhopH2 trafficking appears to be adversely affected (Figure 2).

Figure 2 shows that only RhopH3 is involved in erythrocyte invasion. Because GlcN-treated R2glmS invades normally despite undetectable levels of RhopH2 and CLAG3 (Figure 2), we refute long-standing proposals that these two proteins contribute to invasion. Because RhopH3 traffics normally to rhoptries in this knockdown, R2glmS parasites cannot exclude a RhopH3 contribution. The other knockdown parasite, R3glmS, addresses this unknown and shows a 50% reduction in invasion with 4 mM GlcN. This reduction can be confidently attributed to RhopH3 alone because contributions from the other proteins are excluded by normal invasion in the RhopH2 knockdown.

The late effects of knockdown are now shown separately (new Figure 3). Here, both knockdowns exhibit complete loss of all three proteins in immunoblots and IFAs (Figure 3). These finding indicate that RhopH2 and RhopH3 require interactions with each other for transfer to the new host cell at invasion; loss of CLAG3 is not surprising at this stage because it was already degraded due to early effects in Figure 2.

*Figure 3: This figure again shows the possible combinatorial function and cumulative phenotypes of the 2 knock-downs vs the 3 affected proteins. It should be a manor goal of the revision to address this issue and untangle the phenotypes/functions.*

We agree and have clarified the effects of stage-specific knockdown on trafficking and transfer of the 3 proteins to invaded erythrocytes (addressed above, revised Figure 2 and Figure 3).

The comment here refers to data now presented in Figure 4, which combines kinetic osmotic lysis assays and whole-cell patch-clamp to determine that glucosamine-induced knockdown of either RhopH2 or RhopH3 produces essentially complete loss of PSAC activity on the host membrane.

Figure 4 shows that nearly all cells in either R2glmS or R3glmS are refractory to osmotic lysis in sorbitol at 4 mM glucosamine, but WT cells are not significantly affected. (Sorbitol is a sugar alcohol with uptake primarily via PSAC.) Although the GlcN dose-response profiles for R2glmS and R3glmS appear to differ in this figure, statistical testing excludes significant differences between these parasites at each GlcN concentration, as expected if either knockdown abolishes transfer of all complex members.